



# The design and development of a tuneable and portable radiation source for in situ spectrometer characterisation

Marek Šmíd[1], Geiland Porrovecchio[1], Jiří Tesař[1], Tim Burnitt[2], Luca Egli[3], Julian Grőbner[3], Petr Linduška[1], Martin Staněk[4]

5 [1]Czech Metrology Institute, Brno, 638 00, Czech Republic (Dept. of Optics, Prague)
[2] Principal Optics, Woodley, Berkshire, RG5 4PZ, UK
[3]Physikalisch-Meteorologisches Observatorium Davos / World Radiation Centre, Davos, 7260, Switzerland
[4]Czech Hydrometeorological Institute, Hradec Kralove, 503 11, Czech Republic

10 *Correspondence to*: Marek Šmíd (msmid@cmi.cz)

**Abstract.** For spectroradiometers the characterisation of their wavelength scale and spectral bandwidth underpins substantially the quality of measured data. This characterisation can be performed using metrology-grade tuneable monochromatic sources, which are currently available only in a few laboratories world-wide. Yet in numerous applications only the in-field calibration is a feasible solution. We have designed and developed a Tuneable and Portable radiation Source (TuPS) in the wavelength 15 range from 300 nm to 350 nm for the in-field characterization of Dobson and Brewer spectrometers wavelength scale and slit-function with uncertainties better than 0.02 nm in wavelength and with the bandwidth of emitted radiation smaller than 0,1 nm FWHM. The TuPS is designed such that only minor modifications of its optical system extends/shifts its spectral range towards visible and near-infrared spectral regions and thus expand its application for characterisation of any spectroradiometers in the relevant spectral region of interest.

## 1 Motivation, specific objective

The Dobson and Brewer spectrophotometers are the main instruments used to monitor the ozone layer, even though Dobson spectrophotometers are no longer being manufactured (Dobson, 1968, Kerr et al., 1981). Although each network-type is in itself consistent, total column ozone retrieved from the two instrument types differ by up to 3 %, significantly larger than the consistency of better than ±0.5 % which can be achieved within Brewer or Dobson spectrometers instruments network 25 (Vaniček 2006, 2012). This large discrepancy currently precludes a merging of both datasets and an eventual replacement of one instrument with another type. There is therefore a need for an improved characterization and calibration of the Dobson and Brewer instruments, particularly by involving the reference instruments of each network. The bandwidths and wavelength scale accuracy of the Dobson spectrophotometer are not known for each instrument, but assumed to be equal to the world reference Dobson (Komhyr et al., 1993). Currently tuneable monochromatic sources which could be used for characterisation of 30 Dobsons and Brewers (Redondas et al. 2018) are complex and cumbersome systems that are only found in a few laboratories world-wide and cannot be used for in-field calibrations as requested by this global spectrometers network. Such laboratory–based





characterisations were performed in CMI and PTB (Köhler, et al., 2018) and requested typically a couple of days' time for each spectrometer plus additional time necessary for shipping of often heavy and large devices under test from their permanent in-filed installation down to the metrology laboratory.

This work describes the design and development of a field Tuneable and Portable radiation Source (TuPS) for the wavelength range 300 nm to 350 nm dedicated for an in-field characterization of Dobson spectroradiometers wavelength scale accuracy and slit-function measurement providing the uncertainties of better than 0.02 nm in the wavelength scale and emitting the output radiation with the bandwidth of approximately 0,1 nm and reports on its long-term temporal stability. Moreover the TuPS design was such that only minor modifications of its optical system can extend or shift its spectral range towards visible and near-infrared

spectral region and thus expand its application for spectral characterisation of any spectrometers in demanded spectral region of interest.

## 2 Methods

The TuPS is composed of the combination of a broadband source and an optical tuneable dispersion system; the latter rejects all but a narrow wavelength band, thus rendering the combination of a narrow-band, tuneable source. The dispersion system is optically

similar to a spectrometer but modified to act as a narrow-band tuneable filter for a broadband source. With regards to the spectral bandwidth and the uncertainty of the wavelength scale of emitted radiation the requested parameters are given directly by the spectral dependence of the expected slit functions to be characterized. The values were set to 0.1 nm Full Width at Half Maximum (FWHM) for bandwidth and 0.05 nm OR better in the uncertainty of setting the central wavelength of emitted radiation. Optimal level of total radiant flux emitted from TuPS was found experimentally during the laboratory-based characterization of Dobson

#074 (Köhler et al., 2018) as a value that ranges from 10 nW to 1000 nW for an output beam's spectral bandwidth of 0.1 nm.

The optical layout was designed using the optical simulation tool Zemax®with the aim to design a tuning machine that uses exclusively commercially available off-the-shelf opto-mechanical components and that provides the high spectral accuracy, the narrow bandwidth and the necessary optical power required by the Dobson. A further constraint was on the TuPS physical dimensions requiring it to be being easily transportable such that it could be used for in-field characterization of Dobson

spectrophotometers.

The schematic diagram of the resulting layout of the TuPS dispersion system is shown in Figure 1.

It consists of a 100 μm input pinhole (IP), a 100 μm output slit vertically oriented (OS), two identical off axis-parabolic mirrors (PM1 And PM2) and 3600 grooves/mm grating optimised for a spectral range of interest. Radiation from the input pinhole is collimated by a parabolic mirror and illuminates the grating. The resulting diffracted radiation is focussed by the second parabolic

mirror forming a spectrum across the exit slit. The central output wavelength is controlled by the angle of the grating, and the bandwidth by the width of the exit slit. A very small vertical shift in the image at the exit port is associated with the rotation of the grating. This shift is of no consequence to the subsequent use of the instrument other than that an exit pinhole may block some of the radiation as the image moves. Therefore, a vertical oriented exit slit is used instead. An optical fibre coupled high intensity





broadband UV discharge lamp was used as input radiation source. The system was designed such that the FWHM of emitted

radiation didn't exceed the value of 0.1 nm for whole spectral range of interest.

The optical set up of the first prototype of TuPS is shown in Figure 2.

The TuPS is built on a custom made 400 mm x 400 mm optical board where the grating and the second parabolic mirror positions are fixed; the input pinhole, the first parabolic mirror and the output slit are mounted on high precision micro metric linear stages to provide the fine adjustment needed to compensate for the focal length tolerance of the parabolic mirrors . Both parabolic mirrors

and the grating are mounted on adjustable stages to adjust the mirrors orientation angles for their optical alignment. The motorized rotation stage that sets the grating angle uses a high-resolution encoder with an accuracy of better than 0.001°. The TuPS light engine is enclosed in a light tight housing.

It is worth noting that the TuPS is designed such that only minor modifications of its optical system, involving mainly the selection of a suitable diffraction grating and the employment of an optimal set of order sorting filters at the input aperture side

can extend/shift the spectral range towards the visible and near-infrared spectral regions and thus expand its application for characterisation of any spectroradiometers in the relevant spectral region of interest.

## 3 Results

Optical characterisation of TuPS light engine, both in terms of the central wavelength of its output radiation and the spectral bandwidth, was performed using the CMI reference tuneable monochromatic source consisting of the CMI fiber coupled tuneable

optical parametric oscillator laser facility (OPO). This system offers the tuneable laser operation over the spectral range from 250 nm to 2500 nm whilst the spectral bandwidth of emitted laser beam doesn't exceed the value of 0.03 nm over all investigated spectral range of interest (https://ekspla.com/product/nt242-series-tunable-wavelength-nanosecond-lasers/#specifications ). The OPO tuneability increment is 0.05 nm.

The OPO laser radiation wavelength and its stability is monitored by calibrated wave meter with accuracy better than 0.01 nm

(Balling, et al., 2012). Schematic diagram of the measurement setup is shown in Figure 3.

### 3.1 TuPS wavelength scale

As shown on the diagram, the power of optical radiation emitted from the TuPS output slit is measured by a calibrated 10 mm x 10 mm Si-based photodiode detector in conjunction with a calibrated trans-impedance amplifier (TIA). The measurements are performed via setting the OPO laser wavelength on desired value and  measuring the power of optical radiation throughput

at TuPS output slit while scanning the grating angle in a set of 60 values  around the expected angle with an angular step as low as 0.001°.These measurements were repeatedly performed at wavelength regions ranging from 300 nm to 330 nm with 5 nm step (in black in Figure 4) plus the 6 typical values measured by the Dobson i.e. 305.5 nm, 311.5 nm, 317.5 nm, 325.0 nm, 332.4 nm, 339.5 nm (in red in Figure 4).

For each wavelength the angular position of the peak $A_\lambda$ relative to the wavemeter-corrected wavelength of the calibrating

laser line λ is calculated using the centroid formula Eq. (1):.





$$A_\lambda = \frac{\sum_{i=0}^{i=N} V_i A_i}{\sum_{i=0}^{i=N} V_i} \qquad\qquad (1)$$

where $V_i$ is the signal in Volts measured at the grating angle $A_i$ and $N$ is the number of measured values around each laser line (typically $N$=60). The resulting relationship between the TuPS grating rotation angle and the TuPS output radiation wavelength
at the output slit (OS) is then determined.

Although in general the relation between the rotation angle of diffraction grating and the central wavelength of the radiation emitted from the TuPS output lit doesn't follow a pure linear character, in the relatively narrow Dobson-required spectral range the linear interpolation become more than sufficient approximation. It is evident from Fig. 5 where we report the results of one
particular TuPS wavelength scale calibration process and where the coefficient of determination R-squared reaches the value 0,9999. Consequently, the residual differences between TuPS set wavelength and measured central wavelength of emitted radiation after wavelength scale calibration performed doesn't exceed the value of 0.01 nm over whole spectral range of interest. To investigate its repeatability, we performed series of repeated wavelength scale calibrations. Results confirmed the repeatability of setting of the desired central wavelength scale of TuPS emitted radiation at the value 0.006 nm (k=1).
It is worth noting,, that the TuPS wavelength scale is recalibrated before and after each in-field measurement campaign (as we report below) and based on the calibration results the two linear interpolation parameters readjusted. Potential differences are then accounted as a temporal stability uncertainty contribution into uncertainty budget associated with that in-field calibration campaign.

As for in-field portable equipment its sensitivity to changing ambient condition is crucial, we performed the characterisation
of TuPS temperature dependence. We have provided the calibration of its wavelength scale in four different ambient temperatures ranging from 20 °C to 30 °C. The results have shown small sensitivity of 0,007 nm/°C (k=1) in this temperature range. It can be explained by mechanical symmetry of the TuPS optical setup as well as its compact construction.

As for in-field portable equipment its sensitivity to changing ambient condition is crucial, we performed the characterisation of TuPS temperature dependence. We have provided the calibration of its wavelength scale in four different ambient
temperatures ranging from 20 °C to 30 °C. The results have shown small sensitivity of 0,007 nm/°C (k=1) D3in this temperature range. It can be explained by mechanical symmetry of the TuPS optical setup as well as its compact construction.

### 3.2 TuPS spectral bandwidth

Using the same data set and the linear relationship between grating angle and wavelength it is also possible to easily assess the bandwidth performance of the TuPS. In the Figure 6 is reported the TuPS bandwidth measurement performed at 305 nm using
the OPO as monochromatic source. The measured angular grating rotation of 0.015° corresponds to a spectral bandwidth of 0.12 nm. That presents a sufficient result providing that we take into account that the measured Dobson spectral bandwidth is about 10 to 40 times larger. Design of TuPS optical set up was optimised such as it keeps the bandwidth of the output radiation



close to constant over the specified spectral range. The measurement we provided proved a small increase of 0.01 nm with increasing wavelength ranging from 0.12 nm at 305 nm (Fig.6) to 0.13 nm at 350 nm.

Taking into account the uncertainty of the central wavelength of TuPS emitted radiation (described above) and its bandwidth the uncertainty of the measurement of spectral bandwidth (FWHM) for the Dobson spectrometer can be estimated as high as 0.05 nm.

### 3.3 TuPS optical output power

For the measurement of the optical radiation output power (optical throughput) of the TuPS light engine in the spectral region
of interest we connected the TuPS input fibre in the final operational configuration with a fibre coupled high intensity broadband UV discharge lamp.

A calibrated Si photodiode positioned at TuPS output measured the power of the optical beam. Considering the output beam shape and its divergence angle the position of calibrated detector was set such that its sensitive area was significantly underfilled. The photodiode photocurrent is converted by a calibrated trans-impedance amplifier.

As shown in Figure 7 the measured values are above 25 nW on 0.1 nm FWHM over all spectral range of interest. Based on the data acquired in CMI during the Dobson calibration performed with the CMI monochromator-based facility (Köhler, et al., 2018) the optical power value of 20 nW is sufficiently intense to be detected by the Dobson with a convenient signal to noise ratio.

### 4 TuPS in-field operation

To perform the Dobson characterization, the TuPS tuneable source is placed on top of the spectrometer so that its output is aligned with the Dobson entrance optics with the sun director removed. The optical coupling includes the Dobson entrance diffuser. The alignment between the two instruments is facilitated by the fact that the TuPS is provided with a telescopic cylinder that fits the Dobson entrance optics. The Dobson's photomultiplier current signal is measured by a digital multi meter (DMM) as the voltage drop across a 470 Ohm resistor present in the Dobson photomultiplier board. The Dobson is set to the
highest sensitivity available (usually correspondent to a photomultiplier high voltage of 800V). Potential stray light coming from the surrounding environment is minimized via covering the Dobson top outlets with light insulation fabric.

Typical TuPS in-field measurement arrangement is shown on the Figure 8.

### 5 Comparison of the TuPS in-field calibration and laboratory-based calibration

To perform the comparison of the results of the TuPS in-field calibration of Dobson spectrometer with the laboratory-based
characterisation, we curried out the in-field calibration of Dobson spectrometer #074 on its measuring side in Czech Hydrometeorological Institute in Hradec Kralove, Czech Republic. The laboratory–based characterisation of the same Dobson #074 was performed earlier in CMI (Köhler, et al., 2018). This laboratory-based calibration measurements requested typically

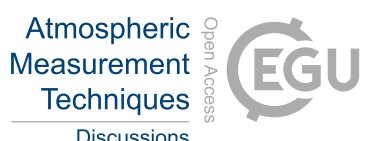

a couple of days' period for calibration itself plus additional time necessary for shipping of heavy and large devices under test from their permanent in-filed installation down to the CMI metrology laboratory, totally typically a period of one week.

Compared to that, the in-field calibration requests approximately 30 minutes time for installation of TuPS system and alignment of TuPs-Dobson coupling, another 30 mins of warm-up time and typically one hour for measuring over all 6 typical bandpasses of the Dobson i.e. 305.5 nm, 311.5 nm, 317.5 nm, 325.0 nm, 332.4 nm, 339.5 nm.

Calibration results are presented in graph shown on Figure 9, where all measured slit functions are shown after normalisation.
The results are as well summarised in Table 1.

The results of comparison of TuPS in-field calibration data and CMI laboratory-based calibration data of Dobson #074 are presented on graph shown on Figure 10.

Despite of a half a year time gap between both measurements the difference between both measurements did not exceed the
value of 0.01 nm in terms of the central wavelength and 0.02 nm in terms of the spectral bandwidth for all 6 Dobson spectral bandpTaking into account the uncertainty asses.

## 6 TuPS temporal stability

Temporal stability of the TuPS light engine was investigated over a period of 2 years from 2017. During the year 2017 the TuPS has participated to five measurement campaigns where it performed the complete characterization of a total of 14 Dobson
spectrometers. Before and after each measurement campaign the TuPS wavelength scale has been recalibrated in CMI laboratory using the OPO laser facility as describe above. The results for calibrations before and after each campaign are reported in Figure 11. The largest differences of both at about 0.025 nm has been recorded after the measurements in AEMET Izana in Spain and the Deutscher Wetterdienst (DWD) in Hohenpeissenberg in Germany campaigns, both over a time interval of approximately 45 days. The TuPS was ground shipped in its protective transportation plastic box in some cases even together
with a number of Dobson spectrometers (for the international Dobson comparison in Izana conducted in September 2017). The different environmental conditions in which the TuPS operated varied from strictly controlled laboratory environments to real in field measurement conditions with temperature as high as 26°C.

To demonstrate the example of the variability of individual Dobsons, we present as an example the results of TuPS-based in-
field characterisation of wavelength scale accuracy and bandwidth of 6 Dobsons performed during the Campaign in El Arenosillo in Spain in September 2017. The results are summarised in Table 2 below.



## 7 Conclusions

The Tuneable and Portable radiation Source (TuPS) was developed as an instrument to be used for determining the slit function and centre wavelength of a Dobson Spectrophotometer. The TuPS was fully characterized at CMI for both bandwidth and the central wavelength accuracy all over the spectral range of interest. Wavelength scale calibration and the investigation of FWHM bandwidth of emitted radiation was performed using the fibre coupled CMI tuneable OPO laser facility in combination with the CMI reference wavemeter and they were proved to be better than 0.02 nm and 0.1 nm respectively. Moreover, the

long term (one year) temporal stability of both key parameters was proved to be better than 0.02 nm while this one-year operation included few in-filed calibrations campaigns abroad involving both shipping and in-field installations.

Considering the achieved results, TuPS-based in-field characterisation might become a prospective method for determination of effective absorption coefficients (EACs) for individual Dobson spectroradiometers of a global monitoring network.


**Acknowledgement**

This work has been supported by the European Metrology Research Programme (EMRP) within the joint research project EMRP ENV59 ATMOZ "Traceability for atmospheric total column ozone". The EMRP is jointly funded by the EMRP participating countries within EURAMET and the European Union.

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




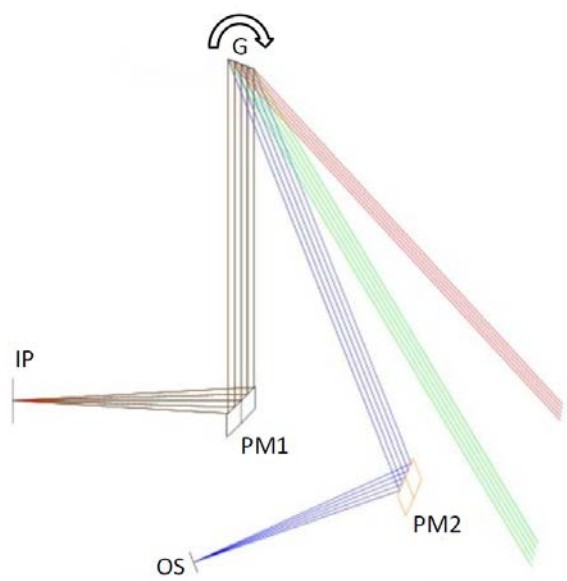

**Figure 1: Schematic diagram of TuPS diffraction system's layout.**

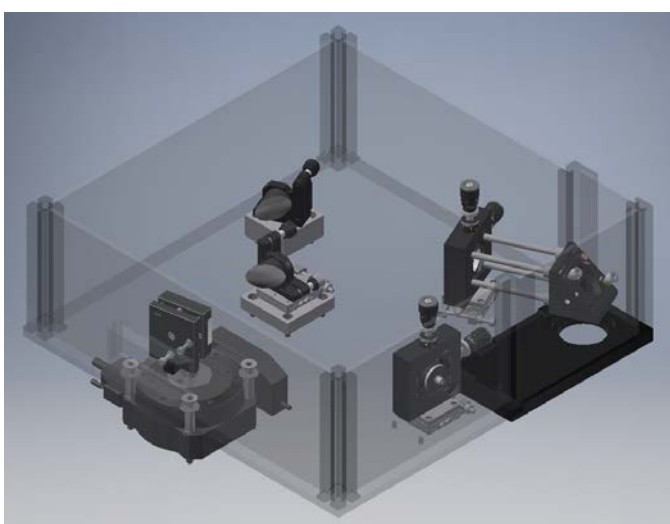

**Figure 2: The optical setup of 1st prototype of TuPS.**





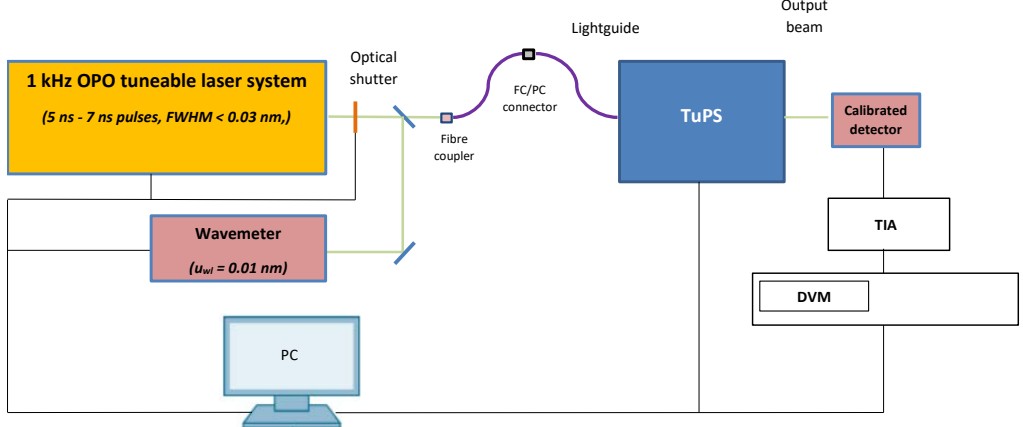


**Figure 3: Schematic diagram of the TuPS characterisation measurement setup.**

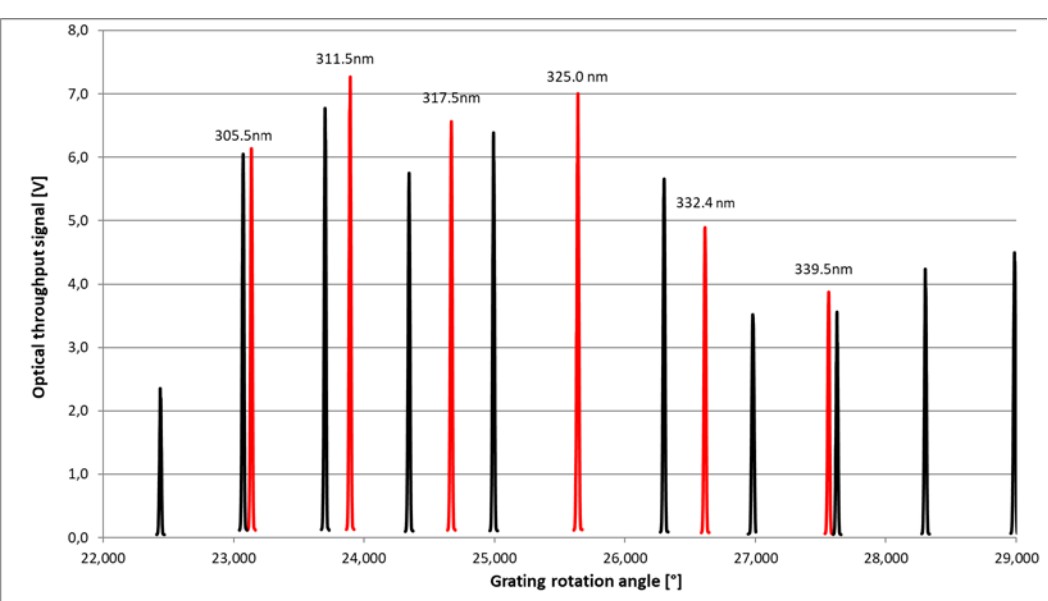

**Figure 4: TuPS wavelength calibration with CMI OPO facility. In red the values of interest for the Dobson spectrometer**



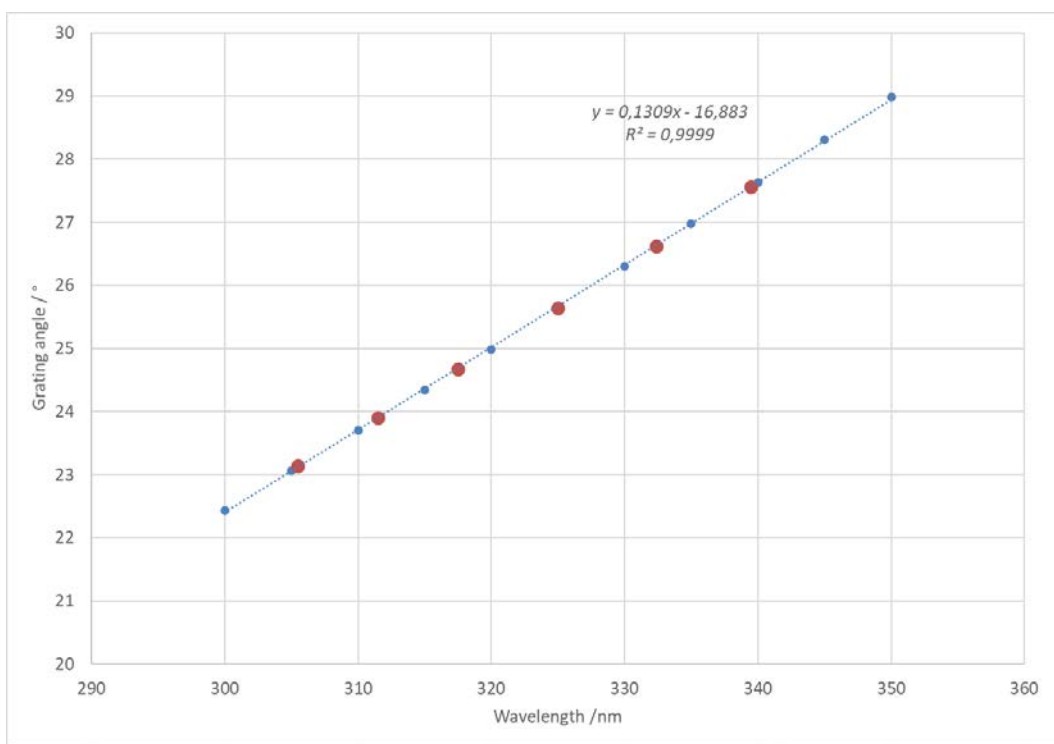

**Figure 5: TuPS wavelength calibration function. Red points represent red values reported in Fig.4**

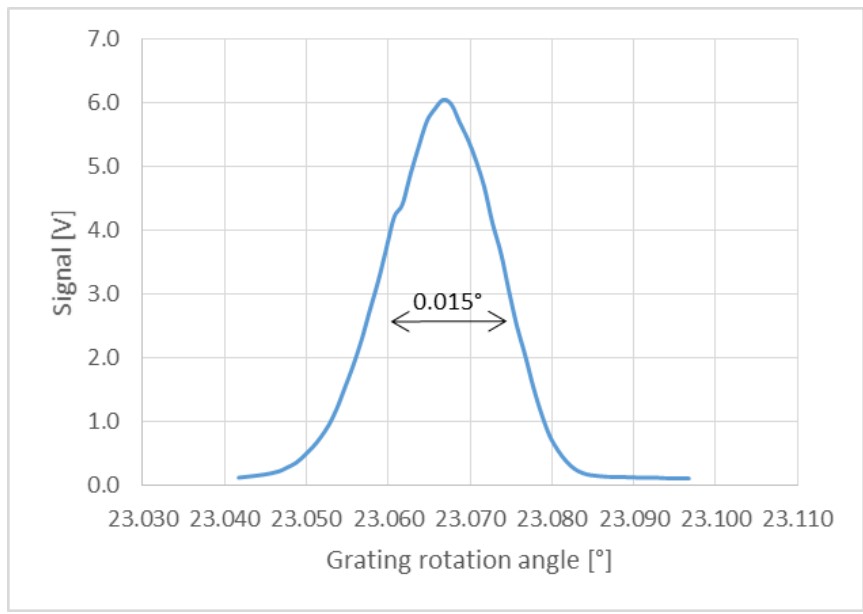

**Figure 6: TuPS bandwidth measurement performed at 305 nm. The width of the pike of 0.015° corresponds to spectral FWHM 0.12 nm**





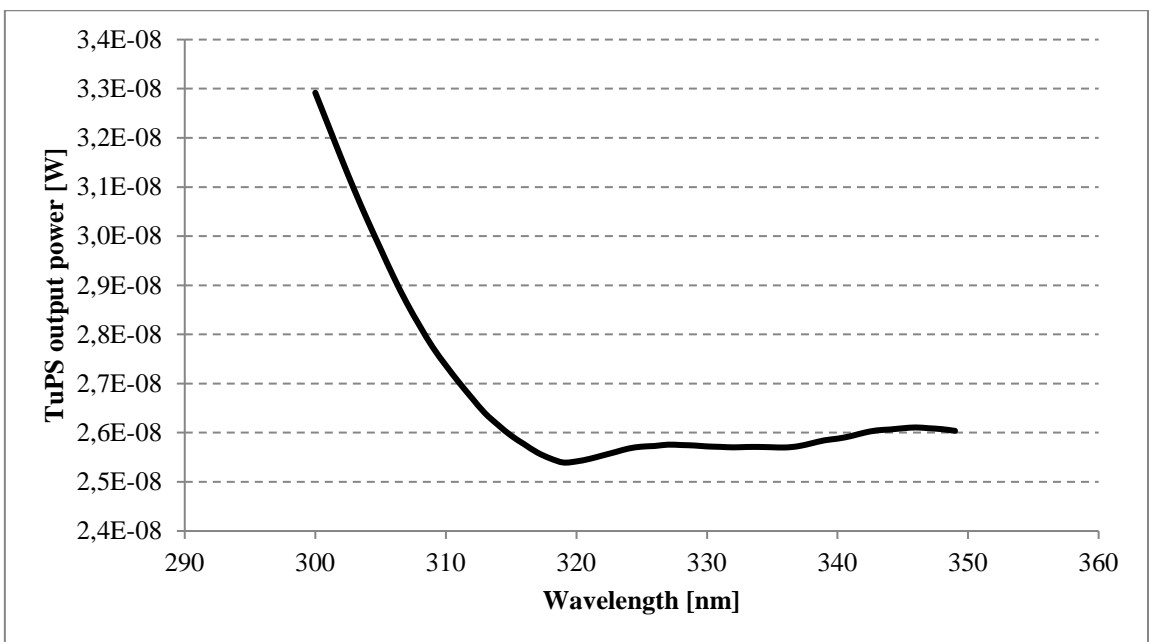

**Figure 7: Optical power output of TuPS light engine in the spectral region of interest.**

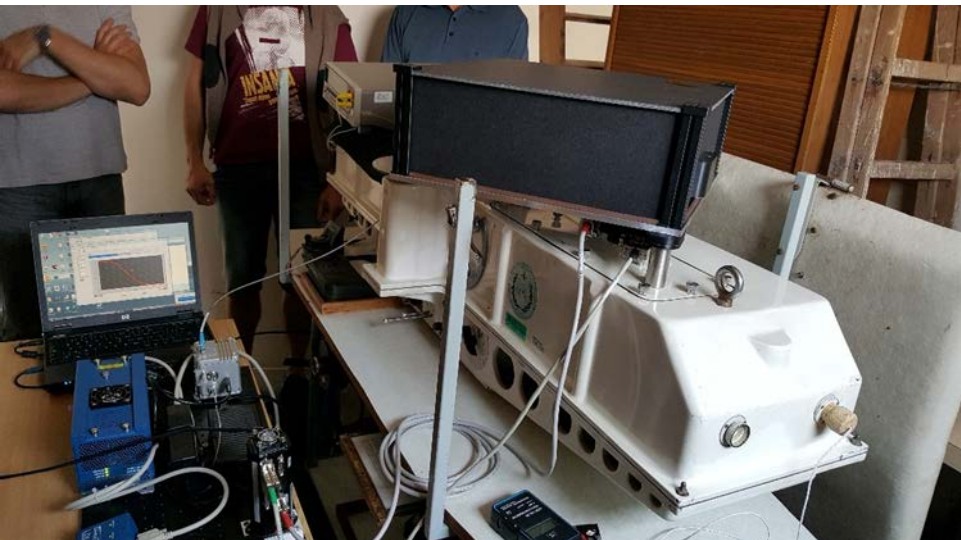

**Figure 8: TuPS in-field measurement arrangement (taken during the Dobson calibration in CHMI Hradec Kralove). On right side - TuPS light engine coupled with the Dobson, on bottom left corner - driving PC and fibre-coupled discharge UV broadband radiation source.**





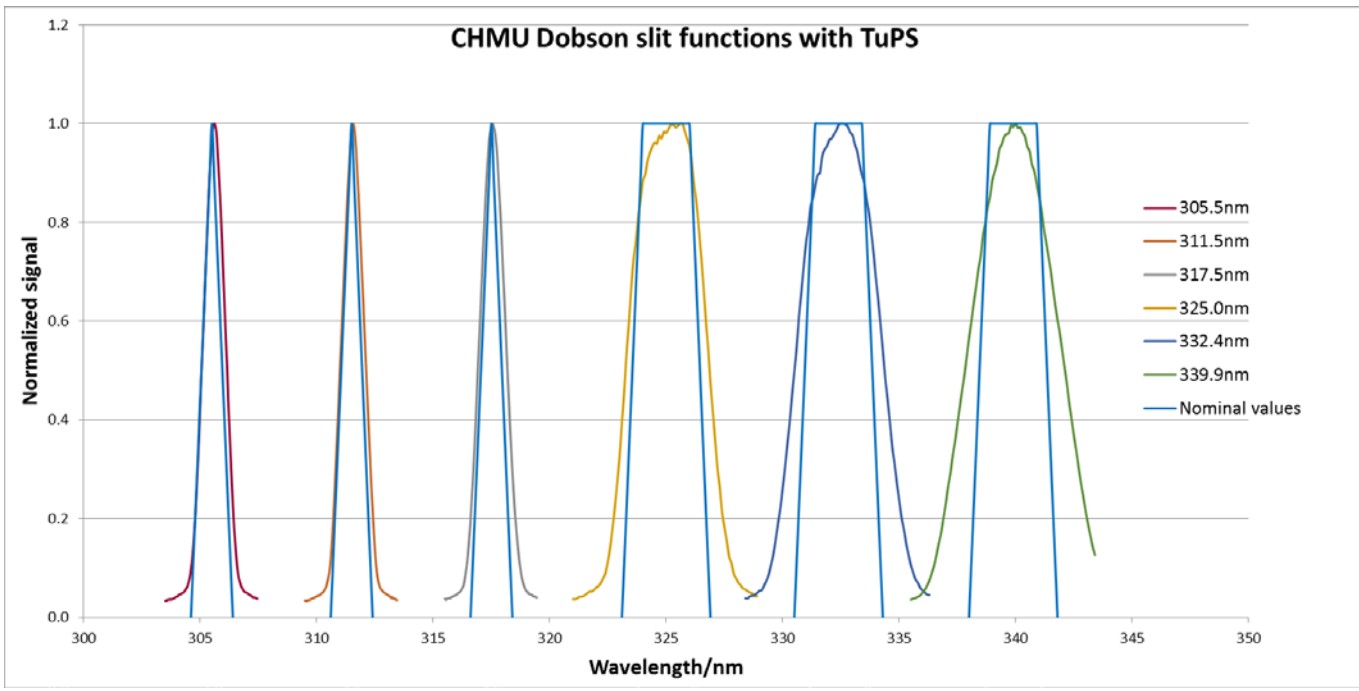


**Figure 9: Dobson #074 slit functions measured during the Tups-based in-field characterisation.**

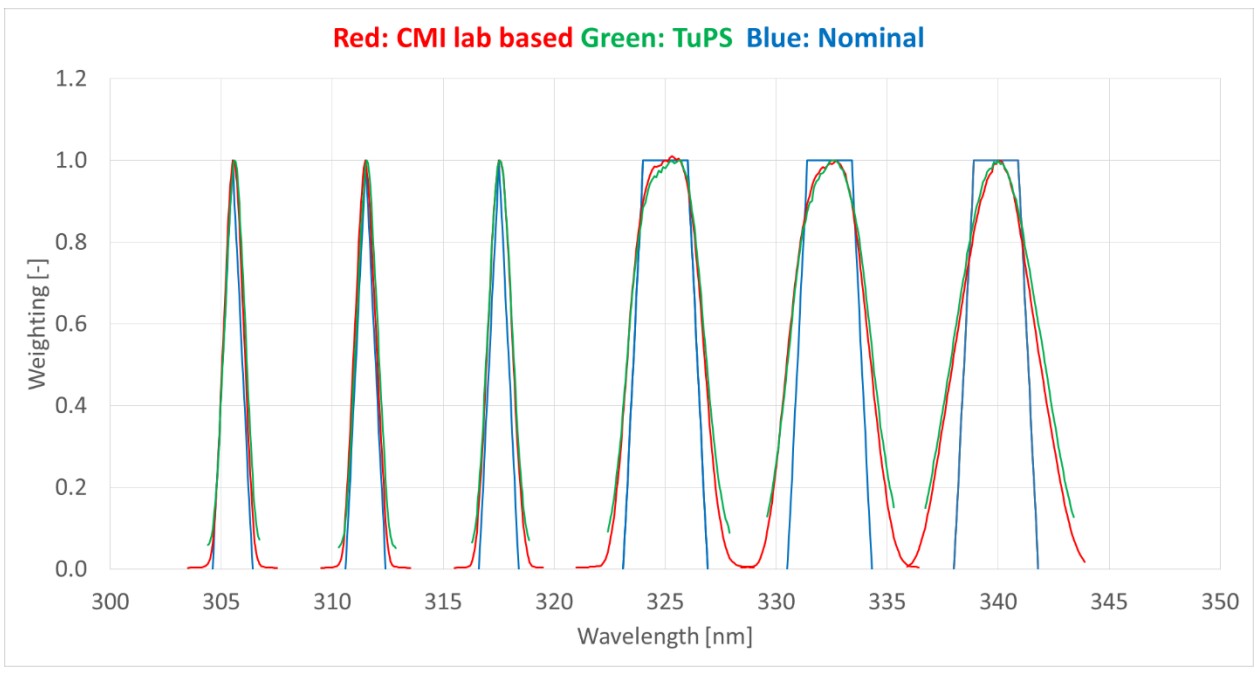

**Figure10: Comparison of in-field calibration and CMI laboratory-based calibration of Dobson #074.**





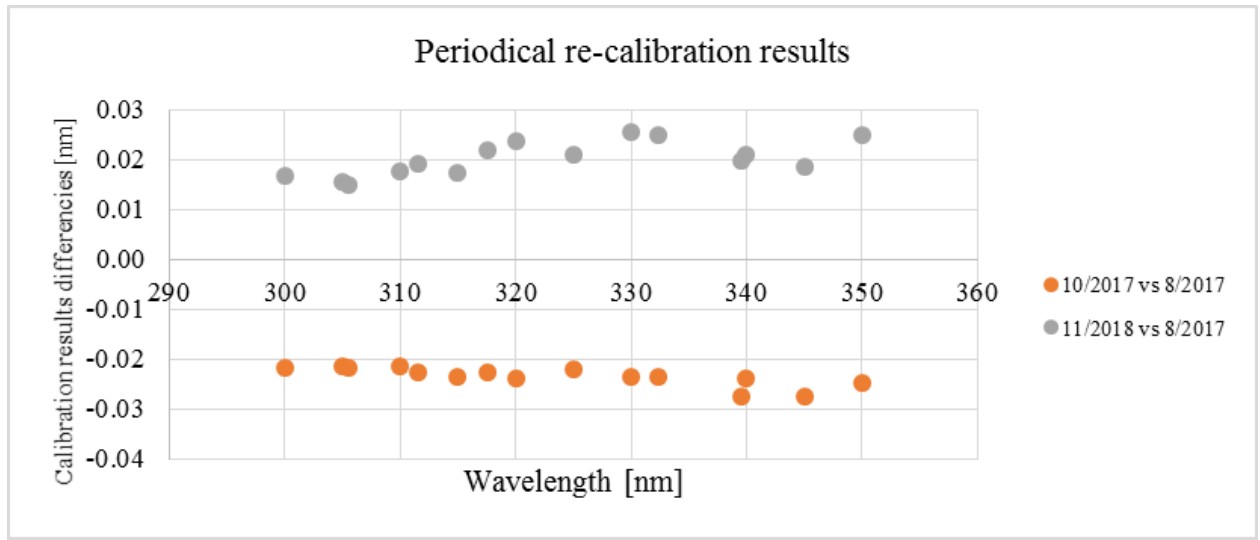

**Figure 11: TuPS results of periodical recalibration of the wavelength scale after each in-field measurement.**






| Peak Nominal [nm] | 305.50 | 311.50 | 317.50 | 325.00 | 332.40 | 339.90 |
|---|---|---|---|---|---|---|
| FWHM Nominal [nm] | 0.90 | 0.90 | 0.90 | 2.90 | 2.90 | 2.90 |
| Peak Measured (center of mass) [nm] | 305.59 | 311.55 | 317.55 | 325.08 | 332.45 | 339.92 |
| FWHM measured [nm] | 1.10 | 1.12 | 1.28 | 3.60 | 3.82 | 4.30 |

**Table 1: Results of the Tups-based in-field characterisation of Dobson  #074**

| Nominal values | Dobson 1 | | Dobson 2 | | Dobson 3 | |
|---|---|---|---|---|---|---|
| Slit/FWHM  [nm] | Peak [nm] | FWHM [nm] | Peak [nm] | FWHM [nm] | Peak [nm] | FWHM [nm] |
| A-S2 (305.5/0.90) | 305.56 | 1.08 | 305.52 | 1.12 | 305.59 | 1.14 |
| C-S2 (311.5/0.90) | 311.55 | 1.14 | 311.56 | 1.14 | 311.97 | 1.25 |
| D-S2 (317.5/0.90) | 317.63 | 1.34 | 317.49 | 1.29 | 317.69 | 1.48 |
| A-S3 (325.0/2.90) | 325.20 | 3.64 | 325.26 | 3.74 | 325.31 | 3.72 |
| C-S3 (332.4/2.90) | 332.49 | 3.93 | 332.57 | 4.04 | 332.51 | 3.98 |
| D-S3 (339.9/2.90) | 340.04 | 4.30 | 340.09 | 4.46 | 340.07 | 4.51 |

| Nominal values | Dobson 4 | | Dobson 5 | | Dobson 6 | |
|---|---|---|---|---|---|---|
| Slit/FWHM  [nm] | Peak [nm] | FWHM [nm] | Peak [nm] | FWHM [nm] | Peak [nm] | FWHM [nm] |
| A-S2 (305.5/0.90) | 305.58 | 1.06 | 305.54 | 1.27 | 305.58 | 1.21 |
| C-S2 (311.5/0.90) | 311.58 | 1.14 | 311.51 | 1.26 | 311.57 | 1.24 |
| D-S2 (317.5/0.90) | 317.63 | 1.33 | 317.54 | 1.41 | 317.63 | 1.34 |
| A-S3 (325.0/2.90) | 325.19 | 3.85 | 325.05 | 3.96 | 325.28 | 3.81 |
| C-S3 (332.4/2.90) | 332.53 | 4.10 | 332.40 | 4.37 | 332.49 | 4.11 |
| D-S3 (339.9/2.90) | 340.03 | 4.47 | 339.88 | 5.02 | 340.02 | 4.53 |


**Table 2: Results of the Tups-based in-field characterisations of different Dobsons which were measured during the Campaign in El Arenosillo in September 2017.**