# Peer review of "The design and development of a tuneable and portable radiation source for in situ spectrometer characterisation"

_Atmospheric Measurement Techniques, 2020_

## Referee Comment (RC1) · Anonymous Referee #1 · 12 Oct 2020

The work describes a portable tunable radiation source for characterization of a UV spectrometer (Dobson and Brewer types) in the field. The operation range is from 300 nm to 350 nm and the claimed uncertainties for the centroid wavelength and spectral bandwidth (FWHM) of the emitted radiation are 0.02 nm and 0.1 nm, respectively. The feasibility of its in-field performance was tested by comparison with the laboratory-based measurement and the temporal stability of the source was verified based on the periodic re-calibrations.

This work provides a practical solution to increase the accuracy of the UV spectrometer network monitoring total column ozone and the achieved performance is satisfactory.

[Figure]

**AMTD**

Therefore, I would like to recommend the publication of this work in AMT as soon as the following comments are considered or clarified:

1. More detailed information in the device design is wished. The design of the developed source is described in Chapter 2 with Fig. 1 and Fig. 2. The authors explained the components of the device but the information is not sufficient to estimate the expected performance. In particular: focal length of the off-axis parabolic mirrors PM1 and PM2, dimension of the grating, f/# or beam divergence, and the specification of the discharge lamp used (lamp type? power?).

2. The uncertainty of the wavelength scale is claimed to be better than 0.02 nm (k = 1 or k = 2 ?) It is however difficult to understand how it was evaluated. In Section 3.1, it states that (page 4, line 106) "the residual differences ... doesn't exceed the value of 0.01 nm over whole spectral range of interest." In addition, repeatability of the wavelength setting is reported to be 0.006 nm (k = 1) and the temperature sensitivity to be 0.007 nm/degC (k = 1). How did these components combined to the final uncertainty of 0.02 nm? When the temperature sensitivity was considerered as an uncertainty component, what was the allowed operation range for the device temperature?

3. The spectral bandwidth of the source is claimed to be "smaller than" 0.1 nm FHMW. However, the measured FWHM in Section 3.2 ranges from 0.12 nm at 305 nm to 0.13 nm at 350 nm, which are all close to but larger than 0.1 nm.

4. From Fig. 6 and Fig. 10, I presume that the source contains some out-of-band stray components. How big is the spectral purity of the developed source? Is it not an relevant specification for testing UV spectrometers?

5. From the result of the temporal stability in Fig. 11, I would say that the change of the scale in a time scale from 8/2017 to 11/2018 is larger than 0.04 nm. This is much larger than the claimed uncertainty of 0.02 nm. Should the long-term instability (including changes due to shipping and in-field environmental conditions) be included in the uncertainty evaluation?

In addition, a few technical corrections are required: - Decimal separator should be consistent. At the current version, points and commas are mixed. - (page 3, line 91) the wavelength range should be from 300 nm to 350 nm instead of 330 nm? - (page 4, line 118) the last paragraph of Section 3.1 is duplicated.
* * *

---

## Short Comment (SC1) · 23 Jan 2021

This paper introduces a portable and accurate method for wavelength calibration for the Dobson spectrophotometers. Wavelength accuracy is indeed and important consideration for instruments used in the long-term ozone observations. The paper is well written and is an important contribution. I have a couple of questions and a few suggestions after reading the paper, the reviewer comments and authors response. First, the paper starts with mentioning both the Brewer and the Dobson spectrophotometers, but eventually discusses only a dedicated Dobson unit. It would be good to know whether the same unit can be used for the Brewers, or a different one is under develop-

ment/testing. Second, and most important: while wavelengths accuracy is crucial and a good calibration unit is always a welcome addition, how big of a problem wavelength accuracy is in the Dobson instruments? The paper results seem to suggest that the central wavelengths are very close to nominal in all tested Dobsons. However, it seems that the results suggest that the slit widths, especially those at longer wavelengths are wider than nominal for all instruments. How does this affect the effective ozone absorption calculations? Here are some cosmetic suggestions: Repetition: lines 114-117 and 118-121 are identical line 15 (and some other places): Both the Brewer and the Dobson are mentioned, but the paper only addresses a dedicated Dobson unit. Suggest wording this line differently to either clarify that the Brewer unit is under construction/testing or to say it's the subject of another paper. Unless the same unit can be used with the Brewer and then this needs to be stated. line 63: "vertically" instead of "vertical" lines 65-71: it is a little confusing when the "second mirror and the grating" are both in the fixed position and are on and adjustable stages. How is it different from the first mirror? line 110: remove double comma line 160: "requires" instead of "requests" line 171: "bandp" - a typo?

---

## Author Comment (AC2) · 1 Feb 2021

Reviewers comment 1 First, the paper starts with mentioning both the Brewer and the Dobson spectrophotometers, but eventually discusses only a dedicated Dobson unit. It would be good to know whether the same unit can be used for the Brewers, or a different one is under development/testing.

The authors response: That is a good comment, thanks for it. The TuPS was designed such that only minor adjustments – optomechanical interface on its entrance optic side - would be needed to extend its application from Dobsons to Brewer spectrometers. This application has although never been requested and thus there has been no further

development of this. As the paper doesn't discuss this in details, it's quite correct to remove the mention of Brewer spectrometer from the Abstract of MS text in line 15.

Reviewers comment 2 Second, and most important: while wavelengths accuracy is crucial and a good calibration unit is always a welcome addition, how big of a problem wavelength accuracy is in the Dobson instruments? The paper results seem to suggest that the central wavelengths are very close to nominal in all tested Dobsons. However, it seems that the results suggest that the slit widths, especially those at longer wave­lengths are wider than nominal for all instruments. How does this affect the effective ozone absorption calculations? The authors response: Again, very good comment. The paper .. Köhler, U., Nevas, S., McConville, G., Evans, R., Smid, M., Stanek, M., Redondas, A., and Schönenborn, F.: Optical characterisation of three reference Dob­sons in the ATMOZ Project – verification of G. M. B. Dobson's original specifications, Atmos. Meas. Tech., 11, 1989–1999, https://doi.org/10.5194/amt-11-1989-2018, 2018. describes in details the impact of the TuPs measurements on the ozone absorption co­efficient calculation and its consequences on the ozone retrieval. To address reviewer's comment, we propose to add the following text including the reference to the paper to line 187 of MS: The impact of the TuPs measurements on the ozone absorption coeffi­cient calculation and its consequences on the ozone retrieval is described in details in (Köhler, et al., 2018).

Editorial (cosmetic) comments/suggestions: Repetition: lines 114-117 and 118-121 are identical - text in lines 118-121 will be removed Line 15 (and some other places): Both the Brewer and the Dobson are mentioned, but the paper only addresses a dedicated Dobson unit. Suggest wording this line differently to either clarify that the Brewer unit is under construction/testing or to say it's the subject of another paper. Unless the same unit can be used with the Brewer and then this needs to be stated - see the response to the comment 1 above line 63: "vertically" instead of "vertical" - agreed, thanks lines 65-71: it is a little confusing when the "second mirror and the grating" are both in the fixed position and are on and adjustable stages. How is it different from the first

mirror? – The text was meant more less to give an overview of the optical alignment. To remove any possible ambiguity in text we propose to simplify the sentence in lines 65-71 as follows: The TuPS is built on a custom made 400 mm x 400 mm optical board where the input pinhole, the first parabolic mirror and the output slit are mounted on high precision micro metric linear stages to provide the fine adjustment needed to . . ...

line 110: remove double comma – agreed line 160: "requires" instead of "requests" – agreed line 171: "bandp" - a typo? – yes, it is typo. This sentence needs editing; following change is proposed: In spite of a half a year gap between the measurements the difference between them did not exceed 0.01 nm, in terms of the central wavelength, and 0.02 nm, in terms of the spectral bandwidth, for all 6 spectral bandpasses of the Dobson.
* * *

---

## Referee Comment (RC2) · Anonymous Referee #3 · 2 Feb 2021

The manuscript by Smid et al. presents a portable near-monochromatic light source for calibration of Dobson spectrophotometers. A description of this "Tuneable and Portable radiation Source" (TuPS) is presented and the manuscript shows convincing performance data. The application of the TuPS for characterization of Dobson spectrophotometers is discussed.

Overall, the manuscript does a nice job in the description of the TuPS, although a few more details of its setup would be helpful (see below). The characterization of the TuPS is generally convincing and the example of application to Dobson instruments is helpful to understand the motivation for the development of the TuPS. What the manuscript is currently missing is to provide the reader with a frame of reference on

how the TuPS improves operation of Dobson spectrophotometers, i.e. how the TuPS will improve the accuracy of ozone column data. The introduction also mentions the disagreement between Dobson and Brewer spectrometers as a motivation for the development of the TuPS. This is not further discussed in the manuscript and the reader will be left wondering if the TuPS does in fact solve this problem. Without discussing these issues, the manuscript lack a clear tie to the science the TuPS is intended to support, reducing its impact and making it less useful to the community.

The topic of the manuscript is suitable to publication in AMT. However, without a more detailed discussion of the improvements the TuPS brings to Dobson spectrophotometers and the consistency between Dobson and Brewer observations, the manuscript lacks scientific relevance. I therefore can recommend publication only after this aspect of the study is added and my more detailed comments below are addressed.

**Detailed comments:**
**Section 1.**

— Information on why a better characterization of the spectral characteristics would improve the consistency between Brewer and Dobson spectrometers would be helpful to better motivate the rest of the manuscript. Are the 3% difference due to difference in how absorption cross sections are derived for each instrument? Is this solely due to an imprecise characterization of the spectral characteristics of the two spectrometers?

— Why is it necessary to develop the TuPS? Could a small commercial monochromator not fulfill the same function? What makes commercial, or previously developed research grade, options unsuitable to solve the characterization problem?

— Since the scientific problem identified here seems to be the comparison of Dobson and Brewer spectrometers, why was the instrument only developed for Dobson systems?

**2. Methods:**

— Line 48. Define 'OR'

— It would be helpful to add a table with details of the optical components, such as diameter and focal lengths of mirrors, size and blaze angle of grating, etc.

— Provide more detail on the motorized rotation stage. What is its angle interval per step, repeatability (precision), etc. How does the stage determine its absolute position, does it have a reference point that is used determine the absolute position at start-up?

— Figure 1 and 2: Add a scale to these figures to provide some sense of the size of the TuPS.

— Throughout the manuscript please ensure to consistently use a decimal point or a decimal comma, but not both.

**3. Results**

— Line 109: What does "(k=1)" mean?

— Line 110: More information on the difference of the calibration before and after in-field use would allow the reader to better assess the stability of the TuPS.

— Line 115: Where is the temperature measured? Is this measurement built into the TuPS or are ambient air temperatures used? How long does it take for the TuPS to stabilize thermally? How is the temperature dependent wavelength scale used?

— Line 117: I don't understand this sentence.

— Lines 118 – 121: This is a repeat of the prior five lines.

— The discussion of temperature dependence also begs the question on the dependence on atmospheric pressure, especially considering that some of the Dobson Instruments are located in mountain observatories.

— Figure 5: Provide the errors of the linear fit as a measure of the precision of the wavelength determination.

— Figure 6 would be easier to understand if plotted vs. wavelength rather than angle. Alternatively, a second x-axis could added.

**5. Comparison of the TuPS in-field calibration**

— Line 155 "curried" should be "carried"

— Figure 9 shows the same data as Figure 10 and can be deleted.

— Line 169: Provide more detail about this comparison. When were the measurement made? Was the Dobson instrument transported in-between the measurements?

— Line 171. I don't understand this sentence

**6. TuPS temporal stability**

— Line 174. This is confusing. The TuPS participated in 5 field experiments and was calibrated before and after each experiment. However, data is only shown for three calibrations in Figure 11.

— Line 177 and Figure 11: Can you explain the difference of wavelength calibration of 0.04 nm?

- Line 181: I do not understand this sentence.

- Lines 184-186: I am not certain what the authors want to convey here. Does this demonstrate that the TuPS is stable or that the Dobson spectrometers are all well calibrated?

- This entire section could use a more detailed description of the various measurements, shipping, calibrations in order to convince the reader that the TuPS is in fact as stable as needed for in-field calibrations. Also, this may be a good place to discuss how the use of the TuPS as an in-field calibration helps in making the Dobson data more accurate.

**7. Conclusion**

- Line 198-199: Elaborate how the TuPS will improve the determination of effective absorption cross sections for Dobson instruments. Did you see a difference between the one currently used and those that would be calculated based on the TuPS measurements? How will this help to decrease the inconsistency with the Brewer spectrometer?

---

## Author Comment (AC3) · 25 Feb 2021

| Referee No. | COMMENTS TO THE AUTHOR(S) | | REPLY FROM THE AUTHOR(S) |
|---|---|---|---|
| Anonymous referee #3 | Overall, the manuscript does a nice job in the description of the TuPS, although a few more details of its setup would be helpful (see below). The characterization of the TuPS is generally convincing and the example of application to Dobson instruments is helpful to understand the motivation for the development of the TuPS. What the manuscript is currently missing is to provide the reader with a frame of reference on how the TuPS improves operation of Dobson spectrophotometers, i.e. how the TuPS will improve the accuracy of ozone column data. The introduction also mentions the disagreement between Dobson and Brewer spectrometers as a motivation for the development of the TuPS. This is not further discussed in the manuscript and the reader will be left wondering if the TuPS does in fact solve this problem. Without discussing these issues, the manuscript lack a clear tie to the science the TuPS is intended to support, reducing its impact and making it less useful to the community. The topic of the manuscript is suitable to publication in AMT. However, without a more detailed | | |
| | 1.1. | Information on why a better characterization of the spectral characteristics would improve the consistency between Brewer and Dobson spectrometers would be helpful to better motivate the rest of the manuscript. Are the 3% difference due to difference in how absorption cross sections are derived for each instrument? Is this solely due to an imprecise characterization of the spectral characteristics of the two spectrometers? | The ozone absorption coefficient of all Dobson spectrophotometers are derived from the primary Dobson, defined as the World reference for the global Dobson network. This ozone absorption coefficient is obtained from the convolution of the line spread functions of each slit with the ozone cross sections at a particular ozone temperature. As shown in Koehler et al (2018), the difference in ozone absorption coefficients using either the nominal value or the ones obtained using the actual measured line spread functions can differ by up to 1%. *Köhler, U., Nevas, S., McConville, G., Evans, R., Smid, M., Stanek, M., Redondas, A., and Schönenborn, F.: Optical characterisation of three reference Dobsons in the ATMOZ Project – verification of G. M. B. Dobson's original specifications, Atmos. Meas. Tech., 11, 1989–1999, https://doi.org/10.5194/amt-11-1989-2018, 2018 .* The 3% difference between the Dobson and Brewer results in part due to the use of a nominal instead of the actually measured line spread functions, as well as the inconsistency in the ozone absorption cross-sections used up to now (Bass & Paur, 1985). *Bass, A. M. and Paur, R. J.: The ultraviolet cross-sections of ozone. I. The measurements, II – Results and temperature dependence, in: Atmospheric ozone; Proceedings of the Quadrennial, 1, 606– 616, 1985.* As shown in Groebner et al., 2021 *(Gröbner, J., Schill, H., Egli, L., and Stübi, R.: Consistency of total column ozone measurements between the Brewer and Dobson spectroradiometers of the LKO Arosa and PMOD/WRC Davos, Atmos. Meas. Tech. Discuss. [preprint], https://doi.org/10.5194/amt-2020-497, in review, 2021.)* the consistency can be significantly improved when using the actually measured line spread functions of a Dobson spectrophotometer, in conjunction with the cross-sections from IUP *(Serdyuchenko, A., Gorshelev, V., Weber, M., Chehade, W., and Burrows, J. P.: High spectral resolution ozone absorption crosssections – Part 2: Temperature dependence, Atmos. Meas. Tech. Discuss., 6, 6613–6643, doi:10.5194/amtd-6-6613-2013, 2013.)* |
| | 1.2. | Why is it necessary to develop the TuPS? Could a small commercial monochromator not fulfill the same function? What makes commercial, or previously developed research grade, options unsuitable to solve the characterization problem? | The measurement of the line spread functions of a Dobson spectrophotometer require the use of a tunable radiation source, that is, a tunable monochromatic source that can be used to scan through the line spread function of a spectroradiometer. This was done with a tunable laser source in the laboratory (Komhyr, W. D., Mateer, C. L., and Hudson, R. D.: Effective BassPaur 1985 Ozone absorption coefficients for use with Dobson ozone spectrophotometer, J. Geophys. Res., 98, 20451–20465, https://doi.org/10.1029/93JD00602, 1993. And more recently Köhler, U., Nevas, S., McConville, G., Evans, R., Smid, M., Stanek, M., Redondas, A., and Schönenborn, F.: Optical characterisation of three reference Dobsons in the ATMOZ Project – verification of G. M. B. Dobson's original specifications, Atmos. Meas. Tech., 11, 1989–1999, https://doi.org/10.5194/amt-11-1989-2018, 2018. However this equipment is complex, cumbersome and requires the Dobson spectroradiometer to be moved to the laboratory. Instead, the TuPS allows the measurement to be performed at the location of the instrument, it takes less than 1 hour of measurements and has therefore nearly no impact on the normal operation of the Dobson spectrophotometer. Before the TuPS development started, we made a survey of the market on availability comertially avalable system meeting the TuPS requirements on its optical parameter and portability. To our best knowledge, there were no commercial monochromators available which could simultaneously meet the TuPS specifications for resolution, wavelength stability and size requirements. And such system is stillnot avalable on the market. |
| | 1.3. | Since the scientific problem identified here seems to be the comparison of Dobson and Brewer spectrometers, why was the instrument only developed for Dobson systems? | In contrast to the Dobson spectrophotometer network, each Brewer spectrophotometer is characterised spectrally at regular intervals using a set of spectral emission lamps (mercury and cadmium as the most common). These measurements are then used to define the line spread functions and thereby to calculate the individual ozone absorption coefficient of the brewer spectrophotometer. It therefore does not require a tunable radiation source like the tups to determine its spectral characteristics. Nevertheless, the methodology employed by the Brewer was validated by measuring the characteristics of one Brewer spectrophotometer belonging to the reference triad of the Regional Brewer Calibration Center-Europe (RBCCE) (Redondas et al., Redondas, A., Nevas, S., Berjón, A., Sildoja, M.-M., León-Luis, S. F., Carreño, V., and Santana-Díaz, D.: Wavelength calibration of Brewer spectrophotometer using a tunable pulsed laser and implications to the Brewer ozone retrieval, Atmos. Meas. Tech., 11, 3759–3768, https://doi.org/10.5194/amt-11-3759-2018, 2018. However the Tups is flexible enough so it could also be applied to the Brewer spectrophotometer if the need would arise. |
| Methods | 2.1. | Line 48. Define 'OR' | Typo. This will be corrected to 'or' |
| | 2.2. | It would be helpful to add a table with details of the optical components, such as diameter and focal lengths of mirrors, size and blaze angle of grating, etc. | Same comment has already been responded to Anonymous referee #1. The information requested by referee will be added in the Chapter 2 . - added in line 57 of manuscript: It consists of a 100 µm input pinhole (IP), a 100 µm output slit vertically oriented (OS), two identical 90° off-axis parabolic mirror 25 mm diameter, 203.2 mm effective focal length (PM1 And PM2) and 3600 grooves/mm grating optimised for a spectral range of interest. Radiation from the input pinhole is collimated by a parabolic mirror and illuminates the grating 25 mm across. The resulting diffracted radiation is focussed by the second parabolic mirror forming a spectrum across the exit slit. The central output wavelength is controlled by the angle of the grating, and the bandwidth by the width of the exit slit. A very small vertical shift in the image at the exit port is associated with the rotation of the grating. This shift is of no consequence to the subsequent use of the instrument other than that an exit pinhole may block some of the radiation as the image moves. Therefore, a vertical oriented exit slit is used instead. The input F/# is F/8.1. The output F/# varies with the wavelength. It ranges from F/11.2 at 300 nm to F/12.8 at 350 nm. The An optical fibre coupled high intensity broadband UV discharge lamp (http://www.energetiq.com/fiber-coupled-laser-driven-lightsource-long-life-compact.php) was used as input radiation source. The system was designed such that the FWHM of emitted radiation didn't exceed the value of 0.1 nm for whole spectral range of interest. |
| | 2.3. | Provide more detail on the motorized rotation stage. What is its angle interval per step, repeatability (precision), etc. How does the stage determine its absolute position, does it have a reference point that is used determine the absolute position at start-up? | The motorized stage is equipped with an high resolution 32-bit relative angular encoder. The absolute position is derived by calibrating the stage to its reference point at the start up. The smallest angular step is limited by the reading noise from the encoder which is in the order of 2e-4 part of a degree. |
| | 2.4. | Figure 1 and 2: Add a scale to these figures to provide some sense of the size of the TuPS. | The scale is going to be mentioned in the text of this chapter. To facolitate the readers comfort better, it will be added to the Fig 1 and Fig 2 description |
| | 2.5. | Throughout the manuscript please ensure to consistently use a decimal point or a decimal comma, but not both. | It will be done |
| Results | 3.1. | Line 109: What does "(k=1)" mean? | The standard uncertainty of measurement has been determined in accordance with JCGM 100:2008 document *(Evaluation of measurement data — Guide to the expression of uncertainty in measurement, BIPM, JCGM, 2008, https://www.bipm.org/utils/common/documents/jcgm/JCGM_100_2008_F.pdf).* The reported expanded uncertainty of measurement is stated as the standard uncertainty of measurement multiplied by the coverage factor k corresponding to a coverage probability of approximately 68 %, which for normal distribution corresponds to a coverage factor k = 1. |
| | 3.2. | Line 110: More information on the difference of the calibration before and after in-field use would allow the reader to better assess the stability of the TuPS. | This information is given in detail in Chapter 6, 'TuPS temporal stability'. Specific reference to chapter 6 will be added to the line 110: It is worth noting that the TuPS wavelength scale is recalibrated before and after each in-field measurement campaign (as reported in chapter 6 below) and the two linear interpolation parameters are readjusted according the calibration results |
| | 3.3. | Line 115: Where is the temperature measured? Is this measurement built into the TuPS or are ambient air temperatures used? How long does it take for the TuPS to stabilize thermally? How is the temperature dependent wavelength scale used? | During the temperature dependence measurement the ambient temperature in the climatic box was measured with a Pt100 temperature sensor. One hour period was give to reach thermal equilibrium of the system before each wavelength scale calibration. During that time ambient temperature kept stable within 0,1 °C. Even though a very small temperature sensitivity of 0,007nm/°C was measured, it can be used as a correction factor for TuPS if the ambient temperature differs significantly from the temperature during the TuPS in-lab calibration. |

| | | Comment | Response |
|---|---|---|---|
| | 3.4. | Line 117: I don't understand this sentence. | It refers to the experimental experience that the mechanical symmetry of optical setup typically exhibits lower temperature sensitivity caused by the thermal expansion. |
| | 3.5. | Lines 118 – 121: This is a repeat of the prior five lines. | It will be corrected |
| | 3.6. | The discussion of temperature dependence also begs the question on the dependence on atmospheric pressure, especially considering that some of the Dobson Instruments are located in mountain observatories. | All grating spectrometers are sensitive to air pressure as they discriminate spoectrally the optical radiation based on the actual wavelength. The changes of the actual wavelength in respect to the vaccum wavelenth is then corrected according the Edlen formula for refractive index of air. See for exapmle web-page: http://emtoolbox.nist.gov/Wavelength/Documentation.asp#EdlenorCiddor.

 These corrections are autamitically aplied in TuPS measurement process as they are for Dobsons too.

 Additional atmospheric pressure sensitivity can be inducesd by pressure contraction of the grating, which we consider as neglagable and dont take it into account. |
| | 3.7. | Figure 5: Provide the errors of the linear fit as a measure of the precision of the wavelength determination. | it will be provided in final version |
| | 3.8. | Figure 6 would be easier to understand if plotted vs. wavelength rather than angle. Alternatively, a second x-axis could be added. | it can be done |
| Comparison | 5.1. | Line 155 "curried" should be "carried" | It will be corrected |
| | 5.2. | Figure 9 shows the same data as Figure 10 and can be deleted. | No, it doesn't show the same data. Fig 9 shows Dobson #074 slit functions measured by the Tups-based in-field characterisation. Fig. 10 shows the results of comparison of in-field TuPS based calibration and CMI laboratory-based calibration of Dobson #074. |
| | 5.3. | Line 169: Provide more detail about this comparison. When were the measurement made? Was the Dobson instrument transported in-between the measurements? | A detailed description of the laboratory based chararcterisation of Dobson #074 is given in the paper (Köhler, et al., 2018), which is referred in line 157. This measurements were performed on CMI in 2016 using primary double-monochromator facility, a complex and combersome system of more than 150 kg in total, which is not portable. For this reason the Dobson itself needed to be treansported to CMI laboratory and all characterisation campaign took five days in total (mentioned in the manuscript in lines 157-160). To be noted that the in-field TuPS based characterisation took place in CHMI Hradec Kralove in Dobson #074 place of measurement in 2017, more than half a year later, in that case without the need of moving the Dobson.
 All this information is given in the paper, only the dates of both measurement was missing and we are going to add it in the lines 157 and 160 as follows:
 *This laboratory-based calibration measurements carried out in 2016 requested typically ….* and
 *Compared to that, the in-field calibration in 2017 requests approximately 30 minutes time for installation 160 of TuPS system …* |
| | 5.4. | Line 171: I don't understand this sentence | It is typo. The sentence will be corrected as follows:
 Despite of a half a year time gap between both measurements the difference between both measurements did not exceed the value of 0.01 nm in terms of the central wavelength and 0.02 nm in terms of the spectral bandwidth for all 6 Dobson spectral bandpasses |
| Temporal stab. | 6.1. | Line 174: This is confusing. The TuPS participated in 5 field experiments and was calibrated before and after each experiment. However, data is only shown for three calibrations in Figure 11. | Good comment, thanks for it. There has never been ambition to demontrate all recalibration datam in this paper, we onlly wanted to present these two 'worst cases', the largest differencies measured in pre- and post- campaign TuPS calibrations in CMI. We submit to reformulate the sentence in line 174 and change the Figure 11 caption as follows:
 Line 174:
 The largest differences of both at about 0.025 nm has been recorded after the measurements in AEMET Izana in Spain and the Deutscher Wetterdienst (DWD) in Hohenpeissenberg in Germany campaigns, both over a time interval of approximately 45 days. The TuPS was ground shipped in its protective transportation plastic box in some cases even together 180 with a number of Dobson spectrometers (for the international Dobson comparison in Izana conducted in September 2017). These two results for calibrations before and after each campaign are reported in Figure 11.

 Figure 11 caption:
 The largest differences measured in pre- and post- campaign TuPS wavelenth scale calibrations in CMI. Dobson comparison campaign in AEMET Izana in Spain (Orange circles) and capaign in he Deutscher Wetterdienst (DWD) in Hohenpeissenberg in Germany (grey circles) |
| | 6.2. | Line 177 and Figure 11: Can you explain the difference of wavelength calibration of 0.04 nm? | The same comment has already been asked and responded/explained to Anonymous referee #1. See below:

 This part is actually explained in the original manuscript in lines 112-115:
 It is worth noting, that the TuPS wavelength scale is recalibrated before and after each in-field measurement campaign (as we report below) and based on the calibration results the two linear interpolation parameters readjusted. Potential differences are then accounted as a temporal stability uncertainty contribution into uncertainty budget associated with that in-field calibration campaign.

 To clarify the text in Chapter 6, we have made following changw of the text in lines 178 -179:
 - Before and after each measurement campaign the TuPS wavelength scale has been recalibrated and re-adjusted in CMI laboratory using the OPO laser facility as describe above (see Chapter 3.1, line 112) |
| | 6.3. | Line 181: I do not understand this sentence. | That means that during the campaigns TuPS was operated either on Dobsons in-field measurement sites equipped by small shed with ambient temperature control - laboratory environment - or on free space placed Dobsons. |
| | 6.4. | Lines 184-186: I am not certain what the authors want to convey here. Does this demonstrate that the TuPS is stable or that the Dobson spectrometers are all well calibrated? | As written in the line 184 it demonstrates the variability of individual Dobsons charaterised by TuPS during one campaign in El Arenosillo in Spain in September 2017 |
| | 6.5. | This entire section could use a more detailed description of the various measurements, shipping, calibrations in order to convince the reader that the TuPS is in fact as stable as needed for in-field calibrations. Also, this may be a good place to discuss how the use of the TuPS as an in-field calibration helps in making the Dobson data more accurate. | A detailed description of the TuPS shipping to and from the in-filed campaigns is given later on in section 6. To highlight that the TuPS is in fact as stable as needed for in-field calibrations, we will add the following text earlier in the chapter:
 lines 173 -180:
 The temporal stability of the TuPS light engine was investigated over a period of 2 years from 2017. During the year 2017 the TuPS has participated to five measurement campaigns where it performed the complete characterization of a total of 14 Dobson spectrometers. The TuPS was ground shipped in its protective transportation plastic box in some case together with a number of Dobson spectrometers (for the international Dobson comparison in Izana conducted in September 2017).
 Before and after each measurement campaign the TuPS wavelength scale has been recalibrated in CMI laboratory using the OPO laser facility as describe above. The results of the calibrations before and after each campaign are reported in Figure 11. The largest differences of about 0.025 nm has been recorded after the measurements in AEMET Izana in Spain and the Deutscher Wetterdienst (DWD) in Hohenpeissenberg in Germany campaigns, both over a time interval of approximately 45 days. |
| Conclusion | 7.1. | Line 198-199: Elaborate how the TuPS will improve the determination of effective absorption cross-sections for Dobson instruments. Did you see a difference between the one currently used and those that would be calculated based on the TuPS measurements? How will this help to decrease the inconsistency with the Brewer spectrometer? | the text of the response to the referee very forst comment will be added to the Conclusion chapter in the line 200 and the list of referencies will be updated as follows:
 Line 200:
 The ozone absorption coefficient of all Dobson spectrophotometers follows the one of the primary Dobson, defined as the World reference for the global Dobson network. This ozone absorption coefficient is obtained from the convolution of the line spread functions of each slit with the ozone cross sections at a particular ozone temperature. As shown in (Koehler et al, 2018), the difference in ozone absorption coefficients using either the nominal value or the ones obtained using the actual measured line spread functions can differ by up to 1%.
 Köhler, U., Nevas, S., McConville, G., Evans, R., Smid, M., Stanek, M., Redondas, A., and Schönenborn, F.: Optical characterisation of three reference Dobsons in the ATMOZ Project – verification of G. M. B. Dobson's original specifications, Atmos. Meas. Tech., 11, 1989–1999, https://doi.org/10.5194/amt-11-1989-2018, 2018.
 The 3% difference between the Dobson and Brewer results in part due to the use of a nominal instead of the actually measured line spread functions, as well as the inconsistency in the ozone absorption cross-sections used up to now (Bass & Paur, 1985).
 Bass, A. M. and Paur, R. J.: The ultraviolet cross-sections of ozone. I. The measurements, II – Results and temperature dependence, in: Atmospheric ozone; Proceedings of the Quadrennial, 1, 606– 616, 1985.
 As shown in Groebner et al., 2021 (Gröbner, J., Schill, H., Egli, L., and Stübi, R.: Consistency of total column ozone measurements between the Brewer and Dobson spectroradiometers of the LKO Arosa and PMOD/WRC Davos, Atmos. Meas. Tech. Discuss. [preprint], https://doi.org/10.5194/amt-2020-497, in review, 2021.) the consistency can be significantly improved when using the actually measured line spread functions of a Dobson spectrophotometer, in conjunction with the cross-sections from IUP (Serdyuchenko, A., Gorshelev, V., Weber, M., Chehade, W., and Burrows, J. P.: High spectral resolution ozone absorption crosssections – Part 2: Temperature dependence, Atmos. Meas. Tech. Discuss., 6, 6613–6643, doi:10.5194/amtd-6-6613-2013, 2013.) |

---

## Author Response (AR1)

| Referee No. | | | COMMENTS TO THE AUTHOR(S) | REPLY FROM THE AUTHOR(S) |
|---|---|---|---|---|
| Anonymous #1 | | | The work describes a portable tunable radiation source for characterization of a UV spectrometer (Dobson and Brewer types) in the field. The operation range is from 300 nm to 350 nm and the claimed uncertainties for the centroid wavelength and spectral bandwidth (FWHM) of the emitted radiation are 0.02 nm and 0.1 nm, respectively. The feasibility of its in-field performance was tested by comparison with the laboratory based measurement and the temporal stability of the source was verified based on the periodic re-calibrations.
 This work provides a practical solution to increase the accuracy of the UV spectrometer network monitoring total column ozone and the achieved performance is satisfactory. Therefore, I would like to recommend the publication of this work in AMT as soon as the following comments are considered or clarified: | |
| Anonymous #1 | 1 | | More detailed information in the device design is wished. The design of the developed source is described in Chapter 2 with Fig. 1 and Fig. 2. The authors explained the components of the device but the information is not sufficient to estimate the expected performance. In particular: focal length of the off-axis parabolic mirrors PM1 and PM2, dimension of the grating, f/# or beam divergence, and the specification of the discharge lamp used (lamp type? power?). | The information requested by referee were added in the Chapter 2 .
 *- added in line 57 of manuscript:*

 It consists of a 100 μm input pinhole (IP), a 100 μm output slit vertically oriented (OS), two identical 90° off-axis parabolic mirror 25 mm diameter, 203.2 mm effective focal length (PM1 And PM2) and 3600 grooves/mm grating optimised for a spectral range of interest. Radiation from the input pinhole is collimated by a parabolic mirror and illuminates the grating 25 mm across. The resulting diffracted radiation is focussed by the second parabolic mirror forming a spectrum across the exit slit. The central output wavelength is controlled by the angle of the grating, and the bandwidth by the width of the exit slit. A very small vertical shift in the image at the exit port is associated with the rotation of the grating. This shift is of no consequence to the subsequent use of the instrument other than that an exit pinhole may block some of the radiation as the image moves. Therefore, a vertical oriented exit slit is used instead. The input F/# is F/8.1. The output F/# varies with the wavelength and it ranges from F/11.2 at 300 nm to F/12.8 at 350 nm. The optical fibre coupled high intensity broadband UV discharge lamp (http://www.energetiq.com/fiber-coupled-laser-driven-lightsource-long-life-compact.php) was used as input radiation source. The system was designed such that the FWHM of emitted radiation didn't exceed the value of 0.1 nm for whole spectral range of interest. |
| Anonymous #1 | 2 | | The uncertainty of the wavelength scale is claimed to be better than 0.02 nm (k = 1 or k = 2 ?) It is however difficult to understand how it was evaluated. In Section 3.1, it states that (page 4, line 106) "the residual differences … doesn't exceed the value of 0.01 nm over whole spectral range of interest." In addition, repeatability of the wavelength setting is reported to be 0.006 nm (k = 1) and the temperature sensitivity to be 0.007 nm/degC (k = 1). How did these components combined to the final uncertainty of 0.02 nm? When the temperature sensitivity was considered as an uncertainty component, what was the allowed operation range for the device temperature? | Thanks for the comment. To clarify the uncertainty evaluation we made following modifications of MS:
*-line 14 of MS:*
*We have designed and developed a Tuneable and Portable radiation Source (TuPS) in the wavelength range from 300 nm to 350 nm for the in-field characterization of Dobson and Brewer spectrometers wavelength scale and slit-function with standard uncertainties better than 0.02 nm in wavelength …*

 *-line 123-124 of manuscript:*
*The evaluation of standard uncertainty of the TuPS wavelength scale is reported in Table 3 in Chapter 6.*

 *-line 175 of manuscript:*
**6 TuPS temporal stability and wavelength scale uncertainty evaluation**

 *-line 186 of manuscript:*
*The evaluation of standard uncertainty of the TuPS wavelength scale is reported in Table 3*

 *-line 280 of manuscript:*
*Table 3 was added to the manuscript* |
| Anonymous #1 | 3 | | The spectral bandwidth of the source is claimed to be "smaller than" 0.1 nm FHMW. However, the measured FWHM in Section 3.2 ranges from 0.12 nm at 305 nm to 0.13 nm at 350 nm, which are all close to but larger than 0.1 nm. | *- text in line 16-17 of manuscript amended:*
*.. with the bandwidth of emitted radiation smaller than 0,13 nm FWHM.* |

| Referee No. | | COMMENTS TO THE AUTHOR(S) | REPLY FROM THE AUTHOR(S) |
|---|---|---|---|
| Anonymous #1 | 4 | From Fig. 6 and Fig. 10, I presume that the source contains some out-of-band stray components. How big is the spectral purity of the developed source? Is it not an relevant specification for testing UV spectrometers? | During the TuPS characterisation measurement the out-of-band stray radiation was measured at the levels lower than 3.5E-4 relative. This value is negligible for the application of Dobson spectrometer characterisation. |
| Anonymous #1 | 5 | From the result of the temporal stability in Fig. 11, I would say that the change of the scale in a time scale from 8/2017 to 11/2018 is larger than 0.04 nm. This is much larger than the claimed uncertainty of 0.02 nm. Should the long-term instability (including changes due to shipping and in-field environmental conditions) be included in the uncertainty evaluation? | *This part is actually explained in the original manuscript in lines 112-115:*

*It is worth noting, that the TuPS wavelength scale is recalibrated before and after each in-field measurement campaign (as we report below) and based on the calibration results the two linear interpolation parameters readjusted. Potential differences are then accounted as a temporal stability uncertainty contribution into uncertainty budget associated with that in-field calibration campaign.*

*To clarify the text in Chapter 6, we have made following change of the text in lines 178 -179:*
*- Before and after each measurement campaign the TuPS wavelength scale has been recalibrated* *and re-adjusted* *in CMI laboratory using the OPO laser facility as describe above* *(see Chapter 3.1, line 112)* |
| **Sevastjuk SC #1** | | This paper introduces a portable and accurate method for wavelength calibration for the Dobson spectrophotometers. Wavelength accuracy is indeed and important consideration for instruments used in the long-term ozone observations. The paper is well written and is an important contribution. I have a couple of questions and a few suggestions after reading the paper, the reviewer comments and authors response.: | |
| Sevastjuk SC #1 | 1 | First, the paper starts with mentioning both the Brewer and the Dobson spectrophotometers, but eventually discusses only a dedicated Dobson unit. It would be good to know whether the same unit can be used for the Brewers, or a different one is under development/testing. | *That is a good comment, thanks for it. The TuPS was designed such that only minor adjustments – optomechanical interface on its entrance optic side - would be needed to extend its application from Dobson to Brewer spectrometers. This application has although never been requested and thus there has been no further development of this.*
*As the paper doesn't discuss this in details, it's probably correct to remove the mention of Brewer spectrometer from the Abstract of MS text in line 15.* |
| Sevastjuk SC #1 | 2 | Second, and most important: while wavelengths accuracy is crucial and a good calibration unit is always a welcome addition, how big of a problem wavelength accuracy is in the Dobson instruments? The paper results seem to suggest that the central wavelengths are very close to nominal in all tested Dobsons. However, it seems that the results suggest that the slit widths, especially those at longer wavelengths are wider than nominal for all instruments. How does this affect the effective ozone absorption calculations? | *This topic is addressed broadly in answers to Anonymous Referee #3, comments 1.1 and 7.1, see below* |
| Sevastjuk SC #1 | 3 | Repetition: lines 114-117 and 118-121 are identical | *text in lines 118-121 removed* |
| Sevastjuk SC #1 | 4 | Line 15 (and some other places): Both the Brewer and the Dobson are mentioned, but the paper only addresses a dedicated Dobson unit. Suggest wording this line differently to either clarify that the Brewer unit is under construction/testing or to say it's the subject of another paper. Unless the same unit can be used with the Brewer and then this needs to be stated | *responded above, see comment 1* |
| Sevastjuk SC #1 | 5 | line 63: "vertically" instead of "vertical" | *in line 63 corrected* |
| Sevastjuk SC #1 | 6 | lines 65-71: it is a little confusing when the "second mirror and the grating" are both in the fixed position and are on and adjustable stages. How is it different from the first mirror? | *The text was meant moreless to give an overview of the optical alignment. To remove any possible ambiguity in text, we propose to simplify the sentence in lines 65-71 as follows:*
*The TuPS is built on a custom made 400 mm x 400 mm optical board where the input pinhole, the first parabolic mirror and the output slit are mounted on high precision micro metric linear stages to provide the fine adjustment needed to …..* |
| Sevastjuk SC #1 | 7 | line 110: remove double comma | *removed* |

| Referee No. | | COMMENTS TO THE AUTHOR(S) | REPLY FROM THE AUTHOR(S) |
|---|---|---|---|
| Sevastjuk SC #1 | 8 | line 160: "requires" instead of "requests" | *done* |
| Sevastjuk SC #1 | 9 | line 171: "bandp" - a typo? | *yes, it is typo.  This sentence was edited, following change is done :*
In spite of a half a year gap between the measurements the difference between them did not exceed  0.01 nm, in terms of the central wavelength, and 0.02 nm, in terms of the spectral bandwidth, for all 6 spectral band passes of the Dobson. |
| **Anonymous referee #3** | | Overall, the manuscript does a nice job in the description of the TuPS, although a few more details of its setup would be helpful (see below). The characterization of the TuPS is generally convincing and the example of application to Dobson instruments is helpful to understand the motivation for the development of the TuPS. What the manuscript is currently missing is to provide the reader with a frame of reference on how the TuPS improves operation of Dobson spectrophotometers, i.e. how the TuPS will improve the accuracy of ozone column data. The introduction also mentions the disagreement between Dobson and Brewer spectrometers as a motivation for the development of the TuPS. This is not further discussed in the manuscript and the reader will be left wondering if the TuPS does in fact solve this problem. Without discussing these issues, the manuscript lack a clear tie to the science the TuPS is intended to support, reducing its impact and making it less useful to the community. The topic of the manuscript is suitable to publication in AMT. However, without a more detailed discussion of the improvements the TuPS brings to Dobson spectrophotometers and the consistency between Dobson and Brewer observations, the manuscript lacks scientific relevance. I therefore can recommend publication only after this aspect of the study is added and my more detailed comments below are addressed. | |
| Anonymous referee #3 | 1.1. | Information on why a better characterization of the spectral characteristics would improve the consistency between Brewer and Dobson spectrometers would be helpful to better motivate the rest of the manuscript. Are the 3% difference due to difference in how absorption cross sections are derived for each instrument? Is this solely due to an imprecise characterization of the spectral characteristics of the two spectrometers? | First, here is the response to the referee:
The ozone absorption coefficient of all Dobson spectrophotometers are derived from the primary Dobson, defined as the World reference for the global Dobson network. This ozone absorption coefficient is obtained from the convolution of the line spread functions of each slit with the ozone cross sections at a particular ozone temperature. As shown in (Koehler et al., 2018), the difference in ozone absorption coefficients using either the nominal value or the ones obtained using the actual measured line spread functions can differ by up to 1%.
The 3% difference between the Dobson and Brewer results in part due to the use of a nominal instead of the actually measured line spread functions, as well as the inconsistency in the ozone absorption cross-sections used up to now (Bass & Paur, 1985).
 As shown in (Groebner et al., 2021), the consistency can be significantly improved from 3% to better than 1 % when using the actually measured line spread functions of a Dobson spectrophotometer, in conjunction with the cross-sections from IUP (Serdyuchenko et al., 2013).
References :
Köhler, U., Nevas, S., McConville, G., Evans, R., Smid, M., Stanek, M., Redondas, A., and Schönenborn, F.: Optical characterisation of three reference Dobson in the ATMOZ Project – verification of G. M. B. Dobson's original specifications, Atmos. Meas. Tech., 11, 1989–1999, https://doi.org/10.5194/amt-11-1989-2018, 2018.
Bass, A. M. and Paur, R. J.: The ultraviolet cross-sections of ozone. I. The measurements, II – Results and temperature dependence, in: Atmospheric ozone; Proceedings of the Quadrennial, 1, 606– 616, 1985.
Gröbner, J., Schill, H., Egli, L., and Stübi, R.: Consistency of total column ozone measurements between the Brewer and Dobson spectroradiometers of the LKO Arosa and PMOD/WRC Davos, Atmos. Meas. Tech. Discuss. [preprint], https://doi.org/10.5194/amt-2020-497, in review, 2021.
Serdyuchenko, A., Gorshelev, V., Weber, M., Chehade, W., and Burrows, J. P.: High spectral resolution ozone absorption cross-sections – Part 2: Temperature dependence, Atmos. Meas. Tech. Discuss., 6, 6613–6643, doi:10.5194/amtd-6-6613-2013, 2013.)
We address it in the MS in 2 parts. Following sentence is added in Line 26, chapter 'Motivation' (plus the reference in relevant part of MS):
*As shown in (Groebner et al., 2021) the consistency can be significantly improved when using the actually measured line spread functions of a Dobson spectrophotometer, in conjunction with the cross-sections from IUP (Serdyuchenko et al., 2013).* |

| Referee No. | | COMMENTS TO THE AUTHOR(S) | REPLY FROM THE AUTHOR(S) |
|---|---|---|---|
| Anonymous referee #3 | 1.2. | Why is it necessary to develop the TuPS? Could a small commercial monochromator not fulfil the same function? What makes commercial, or previously developed research grade, options unsuitable to solve the characterization problem? | First, here is the response to the referee:

The measurement of the line spread functions of a Dobson spectrophotometer require the use of a tunable radiation source, that is a tunable monochromatic source that can be used to scan through the line spread function of a spectroradiometer. This was done with a tunable laser source in the laboratory (Komhyr, W. D., Mateer, C. L., and Hudson, R. D.: Effective BassPaur 1985 Ozone absorption coefficients for use with Dobson ozone spectrophotometer, J. Geophys. Res., 98, 20451–20465, https://doi.org/10.1029/93JD00602, 1993.)
And more recently Köhler, U., Nevas, S., McConville, G., Evans, R., Smid, M., Stanek, M., Redondas, A., and Schönenborn, F.: Optical characterisation of three reference Dobsons in the ATMOZ Project – verification of G. M. B. Dobson's original specifications, Atmos. Meas. Tech., 11, 1989–1999, https://doi.org/10.5194/amt-11-1989-2018, 2018.
However this equipment is complex, cumbersome and requires the Dobson spectroradiometer to be moved to the laboratory. Instead, the TuPS allows the measurement to be performed at the location of the instrument, it takes less than 1 hour of measurements and has therefore nearly no impact on the normal operation of the Dobson spectrophotometer.
Before the TuPS development started, we made a survey of the market on availability commercially available system meeting the TuPS requirements on its optical parameter and portability. To our best knowledge, there were no commercial monochromators available which could simultaneously meet the TuPS specifications for resolution, wavelength stability and size requirements. Such system is still not available on the market.

We believe the chapter 'Motivation' mostly responded the comment as it was, yet to emphasise this part, following of the texts were added in the 'motivation' Chapter:
lines 33-34:
Such laboratory–based characterisations *were performed with a tuneable laser source (Komhyr et al., 1993) and more recently* in CMI and PTB (Köhler, et al., 2018) ...
line 43:
*To our best knowledge, there were no commercial monochromators available which could simultaneously meet the TuPS specifications for resolution, wavelength stability and portable size requirements.* |
| Anonymous referee #3 | 1.3. | Since the scientific problem identified here seems to be the comparison of Dobson and Brewer spectrometers, why was the instrument only developed for Dobson systems? | In contrast to the Dobson spectrophotometer network, each Brewer spectrophotometer is characterised spectrally at regular intervals using a set of spectral emission lamps (mercury and cadmium as the most common). These measurements are then used to define the line spread functions and thereby to calculate the individual ozone absorption coefficient of the brewer spectrophotometer. It therefore does not require a tunable radiation source like the tups to determine its spectral characteristics. Nevertheless, the methodology employed by the Brewer was validated by measuring the characteristics of one Brewer spectrophotometer belonging to the reference triad of the Regional Brewer Calibration Center-Europe (RBCCE) (Redondas et al.,
Redondas, A., Nevas, S., Berjón, A., Sildoja, M.-M., León-Luis, S. F., Carreño, V., and Santana-Díaz, D.: Wavelength calibration of Brewer spectrophotometer using a tunable pulsed laser and implications to the Brewer ozone retrieval, Atmos. Meas. Tech., 11, 3759–3768, https://doi.org/10.5194/amt-11-3759-2018, 2018.
However the TuPS is flexible enough so it could also be applied to the Brewer spectrophotometer if the need would arise. |
| Anonymous referee #3 | 2.1. | Line 48. Define 'OR' | Typo. This will be corrected to 'or' |

TuPS for AMT Responses to Referees 2021-03-23

| Referee No. | | COMMENTS TO THE AUTHOR(S) | REPLY FROM THE AUTHOR(S) |
|---|---|---|---|
| Anonymous referee #3 | 2.2. | It would be helpful to add a table with details of the optical components, such as diameter and focal lengths of mirrors, size and blaze angle of grating, etc. | Same as the comment has already been responded to Anonymous referee #1. The information requested by referee are added in the Chapter 2 . - added in line 60 of manuscript: It consists of a 100 μm input pinhole (IP), a 100 μm output slit vertically oriented (OS), two identical 90° off-axis parabolic mirror 25 mm diameter, 203.2 mm effective focal length (PM1 And PM2) and 3600 grooves/mm grating optimised for a spectral range of interest. Radiation from the input pinhole is collimated by a parabolic mirror and illuminates the grating 25 mm across. The resulting diffracted radiation is focussed by the second parabolic mirror forming a spectrum across the exit slit. The central output wavelength is controlled by the angle of the grating, and the bandwidth by the width of the exit slit. A very small vertical shift in the image at the exit port is associated with the rotation of the grating. This shift is of no consequence to the subsequent use of the instrument other than that an exit pinhole may block some of the radiation as the image moves. Therefore, a vertical oriented exit slit is used instead. The input F/# is F/8.1. The output F/# varies with the wavelength. It ranges from F/11.2 at 300 nm to F/12.8 at 350 nm.  An optical fibre coupled high intensity broadband UV discharge lamp (http://www.energetiq.com/fiber-coupled-laser-driven-lightsource-long-life-compact.php) was used as input radiation source. The system was designed such that the FWHM of emitted radiation didn't exceed the value of 0.1 nm for whole spectral range of interest. |
| Anonymous referee #3 | 2.3. | Provide more detail on the motorized rotation stage. What is its angle interval per step, repeatability (precision), etc. How does the stage determine its absolute position, does it have a reference point that is used determine the absolute position at start-up? | This is the deepest technical detail of TuPS, which we deliberately didn´t include in the MS, particularly because of the TuPS potential future commercialisation. For personal interest of the Referee #3: The motorized stage is equipped with an high resolution 32-bit relative angular encoder.  The absolute position is derived by calibrating the stage to its reference point at the start up.  The smallest angular step is limited by the reading noise from the encoder which is in the order of 2e-4 part of a degree. |
| Anonymous referee #3 | 2.4. | Figure 1 and 2: Add a scale to these figures to provide some sense of the size of the TuPS. | The scale is  mentioned in the text of this chapter. To facilitate the readers comfort better, it will be added to the Fig 2  caption. |
| Anonymous referee #3 | 2.5. | Throughout the manuscript please ensure to consistently use a decimal point or a decimal comma, but not both. | Done |
| Anonymous referee #3 | 3.1. | Line 109: What does "(k=1)" mean? | The standard uncertainty of measurement has been determined in accordance with JCGM 100:2008 document *(Evaluation of measurement data — Guide to the expression of uncertainty in measurement, BIPM, JCGM, 2008, https://www.bipm.org/utils/common/documents/jcgm/JCGM_100_2008_F.pdf)*.  The reported expanded uncertainty of measurement is stated as the standard uncertainty of measurement multiplied by the coverage factor k corresponding to a coverage probability of approximately 68 %, which for normal distribution corresponds to a coverage factor k = 1. No changes in the MS |
| Anonymous referee #3 | 3.2. | Line 110: More information on the difference of the calibration before and after in-field use would allow the reader to better assess the stability of the TuPS. | This information is given in detail in Chapter 6, 'TuPS temporal stability'. Specific reference to chapter 6 will be added to the line 110 to serve better the readers comfort:  It is worth noting that the TuPS wavelength scale is recalibrated before and after each in-field measurement campaign (as reported in chapter 6 below) and  the two linear interpolation parameters are readjusted according the calibration results |
| Anonymous referee #3 | 3.3. | Line 115: Where is the temperature measured? Is this measurement built into the TuPS or are ambient air temperatures used? How long does it take for the TuPS to stabilize thermally? How is the temperature dependent wavelength scale used? | During the temperature dependence measurement the ambient temperature in the climatic box was measured with a Pt100 temperature sensor. One hour period was given to reach thermal equilibrium of the system  before each wavelength scale calibration. During that time ambient temperature kept stable within 0,1 °C.  Even though a very small  temperature sensitivity of 0,007nm/°C was measured, it can be used as a correction factor for TuPS if the ambient temperature differs significantly from the temperature during the TuPS in-lab calibration. No changes in the MS |
| Anonymous referee #3 | 3.4. | Line 117: I don't understand this sentence. | It refers to the experimental experience that the mechanical symmetry of optical setup typically exhibits lower temperature sensitivity caused by the thermal expansion. No changes in the MS |
| Anonymous referee #3 | 3.5. | Lines 118 – 121: This is a repeat of the prior five lines. | It will be corrected,- same as Anonymous Referee #1 |

| Referee No. | | COMMENTS TO THE AUTHOR(S) | REPLY FROM THE AUTHOR(S) |
|---|---|---|---|
| Anonymous referee #3 | 3.6. | The discussion of temperature dependence also begs the question on the dependence on atmospheric pressure, especially considering that some of the Dobson Instruments are located in mountain observatories. | All grating spectrometers are sensitive to air pressure as they discriminate spectrally the optical radiation based on the actual wavelength. The changes of the actual wavelength in respect to the vacuum wavelength is then corrected according the Edlen formula for reflective index of air. See for example web-page: http://emtoolbox.nist.gov/Wavelength/Documentation.asp#EdlenorCiddor. These corrections are automatically applied in TuPS measurement process as they are for Dobsons too. Additional atmospheric pressure sensitivity can be induced by pressure contraction of the grating, which is considered as negligible and is not taken into account.
 *No changes in the MS* |
| Anonymous referee #3 | 3.7. | Figure 5: Provide the errors of the linear fit as a measure of the precision of the wavelength determination. | This does not make a good sense. A compete comprehensive information to the reader is given in the text in line 107: *Consequently, the residual differences between TuPS set wavelength and measured central wavelength of emitted radiation after wavelength scale calibration performed doesn't exceed the value of 0.01 nm over whole spectral range of interest.*

 It is worth reminding, that 0.01 nm (10pm) is the uncertainty of our primary CMI wavelength scale calibration facility, as shown in the manuscript.
 *No changes in the MS* |
| Anonymous referee #3 | 3.8. | Figure 6 would be easier to understand if plotted vs. wavelength rather than angle. Alternatively, a second x-axis could added. | Done |
| Anonymous referee #3 | 5.1. | Line 155 "curried" should be "carried" | Corrected |
| Anonymous referee #3 | 5.2. | Figure 9 shows the same data as Figure 10 and can be deleted. | No, it doesn't show the same data. Fig 9 shows Dobson #074 slit functions measured by the Tups-based in-field characterisation. Fig. 10 shows the results of comparison of in-field TuPS based calibration and CMI laboratory-based calibration of Dobson #074.
 *No changes in the MS* |
| Anonymous referee #3 | 5.3. | Line 169: Provide more detail about this comparison. When were the measurement made? Was the Dobson instrument transported in-between the measurements? | A detailed description of the laboratory based characterisation of Dobson #074 is given in the paper (Köhler, et al., 2018), which is referred in line 157. This measurements were performed on CMI in 2016 using primary double-monochromator facility, a complex and cumbersome system of more than 150 kg in total, which is not portable. For this reason the Dobson itself needed to be transported to CMI laboratory and all characterisation campaign took five days in total (mentioned in the manuscript in lines 157-160). To be noted that the in-field TuPS based characterisation took place in CHMI Hradec Kralove in Dobson #074 place of measurement in 2017, more than half a year later, in that case without the need of moving the Dobson.
 All this information is given in the paper, except the dates of both measurement was missing. We added it in the lines 157 and 160 as follows:
 Line 162: *This laboratory-based calibration measurements carried out in 2016 requested typically ....*
 Line 166: *Compared to that, the in-field calibration in 2017 requests approximately 30 minutes time for installation of TuPS system ...* |
| Anonymous referee #3 | 5.4. | Line 171: I don't understand this sentence | It is typo. The sentence will be corrected as follows: (same comment received from SC#1 and responded)
 *Despite of a half a year time gap between both measurements the difference between both measurements did not exceed the value of 0.01 nm in terms of the central wavelength and 0.02 nm in terms of the spectral bandwidth for all 6 Dobson spectral band passes* |

| Referee No. | | COMMENTS TO THE AUTHOR(S) | REPLY FROM THE AUTHOR(S) |
|---|---|---|---|
| Anonymous referee #3 | 6.1. | Line 174: This is confusing. The TuPS participated in 5 field experiments and was calibrated before and after each experiment. However, data is only shown for three calibrations in Figure 11. | Good comment, thanks for it. There has never been ambition to demonstrate all recalibration data in this paper, we only wanted to present these two 'worst cases', the largest differences measured in pre- and post- campaign TuPS calibrations in CMI. We submit to reformulate the sentence in line 174 and change the Figure 11 caption as follows: Line 177: The largest differences of both at about 0.025 nm has been recorded after the measurements in AEMET Izana in Spain and the Deutscher Wetterdienst (DWD) in Hohenpeissenberg in Germany campaigns, both over a time interval of approximately 45 days. The TuPS was ground shipped in its protective transportation plastic box in some cases even together with a number of Dobson spectrometers (for the international Dobson comparison in Izana conducted in September 2017). These two results for calibrations before and after each campaign are reported in Figure 11.

Figure 11 caption: The largest differences measured in pre- and post- campaign TuPS wavelength scale calibrations in CMI. Dobson comparison campaign in AEMET Izana in Spain (Orange circles) and campaign in the Deutscher Wetterdienst (DWD) in Hohenpeissenberg in Germany (grey circles) |
| Anonymous referee #3 | 6.2. | Line 177 and Figure 11: Can you explain the difference of wavelength calibration of 0.04 nm? | The same comment has already been asked and responded/explained to Anonymous referee #1. See below:

This part is actually explained in the original manuscript in lines 112-115: It is worth noting, that the TuPS wavelength scale is recalibrated before and after each in-field measurement campaign (as we report below) and based on the calibration results the two linear interpolation parameters readjusted. Potential differences are then accounted as a temporal stability uncertainty contribution into uncertainty budget associated with that in-field calibration campaign.

To clarify the text in Chapter 6, we have made following change of the text in lines 178 -179: - Before and after each measurement campaign the TuPS wavelength scale has been recalibrated and re-adjusted in CMI laboratory using the OPO laser facility as describe above (see Chapter 3.1, line 112) |
| Anonymous referee #3 | 6.3. | Line 181: I do not understand this sentence. | That means that during the campaigns TuPS was operated either on Dobsons in-field measurement sites equipped by small shed with ambient temperature control laboratory environment - or on free space placed Dobsons. No changes in the MS |
| Anonymous referee #3 | 6.4. | Lines 184-186: I am not certain what the authors want to convey here. Does this demonstrate that the TuPS is stable or that the Dobson spectrometers are all well calibrated? | As written in the line 184 it demonstrates the variability of individual Dobsons characterised by TuPS during one campaign in El Arenosillo in Spain in September 2017 |
| Anonymous referee #3 | 6.5. | This entire section could use a more detailed description of the various measurements, shipping, calibrations in order to convince the reader that the TuPS is in fact as stable as needed for in-field calibrations. Also, this may be a good place to discuss how the use of the TuPS as an in-field calibration helps in making the Dobson data more accurate. | A detailed description of the TuPS shipping to and from the in-filed campaigns is given later on in section 6. To highlight that the TuPS is in fact as stable as needed for in-field calibrations, we will add the following text earlier in the chapter: lines 178 -184: The temporal stability of the TuPS light engine was investigated over a period of 2 years from 2017. During the year 2017 the TuPS has participated to five measurement campaigns where it performed the complete characterization of a total of 14 Dobson spectrometers. The TuPS was ground shipped in its protective transportation plastic box in some case together with a number of Dobson spectrometers (for the international Dobson comparison in Izana conducted in September 2017). Before and after each measurement campaign the TuPS wavelength scale has been recalibrated in CMI laboratory using the OPO laser facility as describe above. The results of the calibrations before and after each campaign are reported in Figure 11. The largest differences of about 0.025 nm has been recorded after the measurements in AEMET Izana in Spain and the Deutscher Wetterdienst (DWD) in Hohenpeissenberg in Germany campaigns, both over a time interval of approximately 45 days. |

                                                        TuPS for AMT Responses to Referees 2021-03-23

| Referee No. | | COMMENTS TO THE AUTHOR(S) | REPLY FROM THE AUTHOR(S) |
|---|---|---|---|
| Anonymous referee #3 | 7.1. | Line 198-199: Elaborate how the TuPS will improve the determination of effective absorption cross sections for Dobson instruments. Did you see a difference between the one currently used and those that would be calculated based on the TuPS measurements? How will this help to decrease the inconsistency with the Brewer spectrometer? | To address this comment, following text is added to the line 207: *The ozone absorption coefficient of all Dobson spectrophotometers follows the one of the primary Dobson, defined as the World reference for the global Dobson network. This ozone absorption coefficient is obtained from the convolution of the line spread functions of each slit with the ozone cross sections at a particular ozone temperature. As shown in (Koehler et al, 2018), the difference in ozone absorption coefficients using either the nominal value or the ones obtained using the actual measured line spread functions can differ by up to 1%.* *The 3% difference between the Dobson and Brewer results in part due to the use of a nominal instead of the actually measured line spread functions, as well as the inconsistency in the ozone absorption cross-sections used up to now (Bass & Paur, 1985). As shown in (Groebner et al., 2021), the consistency can be significantly improved from 3% to better than 1 % when using the actually measured line spread functions of a Dobson spectrophotometer, in conjunction with the cross-sections from IUP (Serdyuchenko et al., 2013).* Following references were added to relevant part of the MS: *Bass, A. M. and Paur, R. J.: The ultraviolet cross-sections of ozone. I. The measurements, II – Results and temperature dependence, in: Atmospheric ozone; Proceedings of the Quadrennial, 1, 606– 616, 1985.* *Gröbner, J., Schill, H., Egli, L., and Stübi, R.: Consistency of total column ozone measurements between the Brewer and Dobson spectroradiometers of the LKO Arosa and PMOD/WRC Davos, Atmos. Meas. Tech. Discuss. [preprint], https://doi.org/10.5194/amt-2020-497, in review, 2021* *Serdyuchenko, A., Gorshelev, V., Weber, M., Chehade, W., and Burrows, J. P.: High spectral resolution ozone absorption cross-sections – Part 2: Temperature dependence, Atmos. Meas. Tech. Discuss., 6, 6613–6643, doi:10.5194/amtd-6-6613-2013, 2013.* |